# TIAM-1 differentially regulates dendritic and axonal microtubule organization in patterning neuronal development through its multiple domains

**Chih-Hsien Lin**[1], **Ying-Chun Chen**[1], **Shih-Peng Chan**[2], **Chan-Yen Ou**[1]*

**1** Institute of Biochemistry and Molecular Biology, College of Medicine, National Taiwan University, Taipei, Taiwan, **2** Graduate Institute of Microbiology, College of Medicine, National Taiwan University, Taipei, Taiwan

* chanyen@ntu.edu.tw

**Data Availability Statement:** All relevant data are within the manuscript and its Supporting Information files.

## Abstract

Axon and dendrite development require the cooperation of actin and microtubule cytoskeletons. Microtubules form a well-organized network to direct polarized trafficking and support neuronal processes formation with distinct actin structures. However, it is largely unknown how cytoskeleton regulators differentially regulate microtubule organization in axon and dendrite development. Here, we characterize the role of actin regulators in axon and dendrite development and show that the RacGEF TIAM-1 regulates dendritic patterns through its N-terminal domains and suppresses axon growth through its C-terminal domains. TIAM-1 maintains plus-end-out microtubule orientation in posterior dendrites and prevents the accumulation of microtubules in the axon. In somatodendritic regions, TIAM-1 interacts with UNC-119 and stabilizes the organization between actin filaments and microtubules. UNC-119 is required for TIAM-1 to control axon growth, and its expression levels determine axon length. Taken together, TIAM-1 regulates neuronal microtubule organization and patterns axon and dendrite development respectively through its different domains.

## Author summary

In neurons, microtubules form an acentrosomal organization to support polarized trafficking and development of neuronal processes. While dendritic branches and the axon grow differentially, it is largely unknown how microtubules organize with actin architectures to promote distinct axon and dendrite development. In this work, we show that a multi-domain RacGEF protein TIAM-1 controls dendrite patterning by its N-terminal domains and suppresses axon development by its C-terminal GEF domains. TIAM-1 is required for maintaining microtubule polarity in dendritic regions and suppressing accumulation of axonal cargos and microtubules. In somatodendritic regions, TIAM-1 binds UNC-119 and maintains the organization between actin and microtubule. TIAM-1 prevents axonal distribution of UNC-119 and thus controls axon development. This work

**Funding:** This study is supported by the grant: MOST 111-2311-B-002-010 to C.-Y. O. from Ministry of Science and Technology, Taiwan. The funders had no role in study design, data collection and analysis, decision to publish, or preparation of the manuscript.

**Competing interests:** The authors have declared that no competing interests exist.

demonstrates how dendritic and axonal microtubule organizations are differentially controlled by TIAM-1.

## Introduction

In neurons, actin filaments and microtubules (MTs) are fundamental frameworks for movement, subcellular compartmentalization, and outgrowth of neurites such as dendrites and axons [1]. Axonal outgrowth is influenced by many layers of regulation, including the production of membrane and cytoplasmic components such as vesicles and cytoskeletons, the transport of these membrane components and cytoskeletons, and the dynamics of the growth cone responding to local chemical cues and adhesion choices [2,3]. In the growth cone, actin filaments form filopodia and lamellipodia in the peripheral region surrounding the central microtubule bundles [4–6]. Actin and microtubule dynamics as well as the interplay between these two cytoskeletons are essential for growth cone behaviors [1,7–9].

The regulation of MT dynamics and stability drive early neurite formation and influence neuronal polarity [8,10]. Polarized MT configuration is the basis for motor-mediated trafficking of axon and dendrite specific cargos [11]. As main tracks for motors, microtubules orient uniformly in plus-end-out direction in axons and often in minus-end-out direction in dendrites of invertebrates but exhibit mixed polarity in vertebrate dendrites [12]. Other than orientation, MTs are diversified by different compositions of tubulin isotypes and post-translational modifications such as tyrosination, polyglutamylation, or acetylation which affect the interaction between motors and MTs [13]. However, how these different MTs assemble into a well-organized network in neurons remains largely unknown.

MT-interacting proteins and motors maintain MT organization in neurons [14,15]. The minus-end-directed motor dynein keeps uniform MT polarity in axons by removing minus-end-out MTs in *Drosophila* [16,17]. The plus-end-directed motor Kinesin-1 mediates the transport of dendritic growth cone microtubule-organizing center (MTOC) and regulates dendritic MT orientation in *C. elegans* [18,19], while other kinesin motors like mitotic Kinesin-6 and -12 as well as Kinesin-2 also regulate MT orientation in dendrites [20–22]. A MT-interacting protein TRIM46 is required to construct the uniform axonal MT bundles in mammalian neurons [23]. Recently, we find that MT bundles are organized by UNC-33/CRMP to form a high-order spindle-like network in the cell body with extensions to axons and dendrites in worm neurons for efficient mitochondrial transport [24]. UNC-33 interacts with UNC-119 and UNC-44/ankyrin to form a membrane-associated complex that anchors MTs to the cortex [25].

Actin filaments cooperate with MTs in many aspects of neuronal development and constitute various specific axonal and dendritic structures [1]. Other than being the primary component in growth cones that are important for axon and dendrite growth, actin filaments form repetitive rings with intercalating spectrin as cortical lattices preferentially in the axonal cortex [26–28]. These periodic actin structures are essential for maintaining axonal microtubule bundles and supporting the neurite shape [29]. In soma and dendritic regions, actin filaments are organized as longitudinal fibers, actin patches, mobile actin blobs, and branched networks in terminal branches and dendritic spines [30–32]. Actin filaments are regulated by protein factors, including nucleating complexes (Arp2/3, WASP, WAVE, Formin), polymerization factors (Profilin, Formin), depolymerization factor (Cofilin/ADF), and Rho family GTPases [33]. These factors regulate the formation of different actin-based structures [29,34,35]. They participate in neurite outgrowth [36–39], neuronal polarity establishment [39–41], and dendritic

spine development [36,42,43]. However, specific roles of these actin regulators between axon and dendrite development remain to be characterized.

In this study, we characterize the roles of actin regulators in the polarized development of axon and dendrite, including ACT-3 (Actin), PFN-1 (Profilin), UNC-60 (Cofilin/ADF), FHOD-1 (Formin), WASP, WAVE, MIG-2 (RhoG), and TIAM-1 (RacGEF). We find that UNC-60 (Cofilin/ADF) dedicatedly suppresses axon growth, and WAVE specifically promotes dendritic arbor growth but not axon growth. Among these regulators, interestingly, TIAM-1 supports dendrite growth while suppresses axon growth. In previous studies, TIAM-1 was reported to regulate neuronal protrusion and dendrite arborization [44,45]. Although TIAM-1 is a GEF factor, its GEF activity is not required for normal dendritic development [46]. Instead, a recent study showed the GEF activity of TIAM-1 is required for dendritic regeneration after laser ablation [47]. Here, we further show that TIAM-1 patterns neuronal development through its different domains: N-terminal EVH1 domain, middle PDZ domain, C-terminal GEF domain. Its N-terminal EVH1 domain regulates high-order dendritic branches; its middle PDZ domain is required for primary dendrite development; its C-terminal GEF domain controls axon outgrowth. TIAM-1 interacts with UNC-119 and regulates neuronal actin and microtubule organization. TIAM-1 restricts axonal distribution of UNC-119, which is necessary for controlling axonal outgrowth.

## Results

### Actin regulators have distinctive roles in axonal growth and dendritic morphogenesis

To identify specific regulators for axonal or dendritic development, we examined the morphogenesis of sensory PVD neurons in putative null mutant alleles of actin regulators, including *mig-2(mu28)* [48], *unc-60(su158)* [49], *wve-1(ok3308)* [50], *pfn-1(ok808)*, *fhod-1 (tm2363)* [51], *wsp-1(gm324)* [52], and *tiam-1(ok772)* [46]. A PVD neuron extends an anterior primary dendrite toward the head, a posterior primary dendrite toward the tail, and an axon that innervates the ventral nerve cord. Primary dendrites develop menorah-like side branches (high-order dendrites) to cover the worm body (Fig 1A and 1B). Both axonal and dendritic processes showed excessive growth in *act-3* (actin) gain-of-function allele *st15* (Fig 1C and 1H–1I), suggesting that actin supports both axonal and dendritic growth. The RhoG putative null mutant *mig-2(mu28)* showed significantly reduced growth of both axonal and dendritic processes (Fig 1D and 1H–1I), indicating that it is also essential for both axonal and dendritic growth.

However, mutants of *wsp-1* (WASP) and *pfn-1* (Profilin) showed no significant change in axonal and dendritic growth (S1 Fig). The *fhod-1* (Formin) mutant had a mild defect (10% decrease) in dendritic growth (S1 Fig). In contrast, the *unc-60* (Cofilin/ADF) mutant (*su158*) displayed axonal overgrowth (with 29% increase) but no increase in total dendrite length (Fig 1E and 1H–1I). Interestingly, *wve-1* (WAVE) mutant (*ok3308*) showed 32% decrease of total dendritic length but no obvious change in axonal growth (Fig 1F and 1H–1I). Therefore, while actin supports both axonal and dendritic processes, specific actin regulators could preferentially promote axonal growth or dendritic arbor development. Different from dedicated axonal or dendritic regulators, we found that the RacGEF TIAM-1 regulates dendritic development and axonal growth in an opposite manner; the *tiam-1* (*ok772*) mutant showed significant axonal outgrowth with about a two-fold increase in length but severely defective dendritic development (Fig 1G–1I). This interesting bimodal phenotype suggests that TIAM-1 likely regulates axonal and dendritic development through different mechanisms.

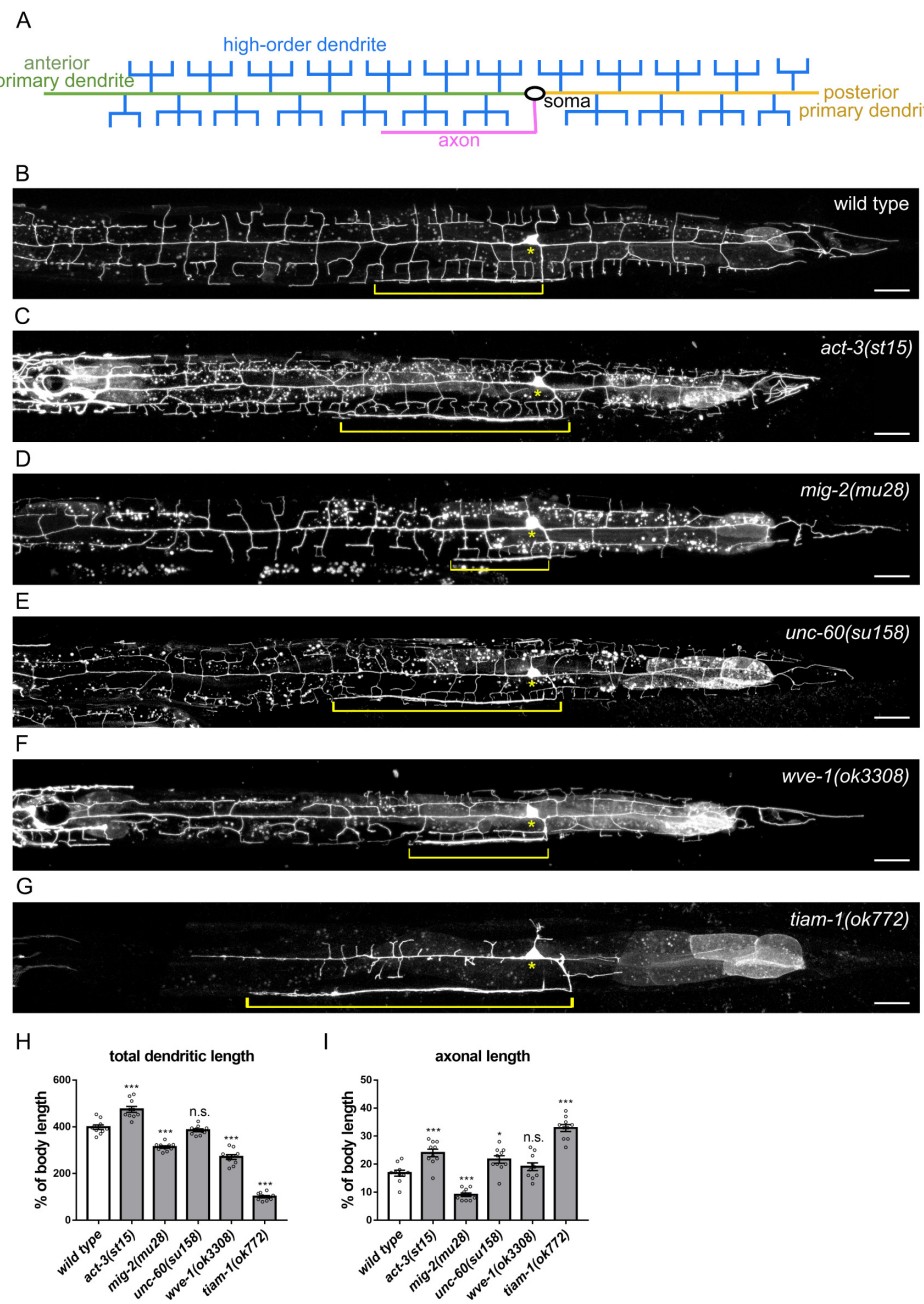

**Fig 1. Actin regulators have distinct roles in axonal and dendritic morphogenesis.** (A) Diagram of PVD neuron morphology. (B-G) PVD neuron expressing myr::mCherry under *Pdes-2* promoter (*ntuIs1*) in wild-type (B), *act-3 (st15)* (C), *mig-2(mu28)* (D), *unc-60(su158)* (E), *wve-1(ok3308)* (F), and *tiam-1(ok772)* (G). Brackets indicate the axons and asterisks indicate the cell body. Scale bar, 20 μm. All worm images were oriented with the anterior toward left and the ventral toward down in this and the following Figures. (H and I) Quantification of total dendritic length (H) and axonal length (I) relative to the body length in wild type and indicated mutants. Error bar represents SEM. n = 10. One-way ANOVA test. n.s., not significant, *P<0.05, **P<0.01, ***P<0.001, compared to wild type.

## Different regions of TIAM-1 regulate the development of distinct neuronal subcompartments

Based on sequence analysis of cDNA clones, *tiam-1* has three transcripts, *tiam-1a*, *tiam-1b*, and *tiam-1c* [44] (Fig 2A). We edited C7 codon of *tiam-1a* into a stop codon to generate the *ntu22* mutant allele by CRISPR-Cas9. Furthermore, we generated the *ntu14* mutant allele by editing both the C7 codon of *tiam-1a* and the W7 codon of *tiam-1b* into stop codons (Fig 2A). *ntu22* showed a reduction of high-order dendritic branches but no defects in either axonal growth or primary dendrite development (S2A and S2G–S2I Fig), indicating the role of *tiam-1a* in regulating high-order dendrite development. In the *ntu14* mutant, not only high-order dendrite formation was severely disrupted, but also the length of primary dendrite reduced, suggesting that *tiam-1b* is further required for primary dendrite development (S2B and S2G–S2I Fig). Since the mutant phenotype of *ntu14* has no significant difference to the putative null allele *ok772* that has a deletion that disrupts all three transcripts [44], *tiam-1a* and *tiam-1b* could be the main transcripts responsible for PVD development.

To study the functions of different domains, we generated extrachromosomal arrays of three transgenes that express TIAM-1A, TIAM-1B, or TIAM-1C, respectively and assayed their activities in *ok772 mutant* (Fig 2A). The full-length form TIAM-1A contains an N-terminal EVH1 domain, a middle region containing a PDZ domain, and a C-terminal DHPH GEF domain. Compared to TIAM-1A, the TIAM-1B lacks the N-terminal EVH1 domain, and TIAM-1C further loses a part of the middle region, containing a truncated PDZ domain and a DHPH GEF domain at the C-terminal region (Fig 2A). While TIAM-1A expression effectively restored both axonal and dendritic growth defects, TIAM-1B expression did not rescue distal high-order dendrite morphogenesis defects, indicating the function of the EVH1 domain in high-order dendrite development (Fig 2B–2C and 2F–2H). In contrast, TIAM-1C expression did not rescue dendritic growth defects but significantly suppressed the axonal overgrowth defect (Fig 2D and 2F–2H). These results suggested that each region of TIAM-1 may be responsible for the development of different neuronal subcompartments; the EVH1 domain is required for the development of high-order dendrites, the PDZ-domain-containing middle region is necessary for primary dendrite development (comparing TIAM-1B with TIAM-1C), and the C-terminal region with the DHPH domain is required for axonal growth (Fig 2I). To test whether axonal growth and dendritic development are respectively controlled by different regions of TIAM-1, we expressed the C-terminal truncated form TIAM-1N539 in *ok772*. Indeed, the expression of TIAM-1N539 restored dendritic development but failed to rescue the axonal outgrowth defect (Fig 2E–2H).

We further introduced different isoforms into *tiam-1* mutant by the single-copy insertion technique that integrates transgene into the same locus of the chromosome to ensure consistent expression of isoforms [53]. These single-copy transgenic strains showed a stronger but similar rescuing effect to extrachromosomal arrays (S2C–S2I Fig). Notably, while TIAM-1N539 and TIAM-1C restored dendrite and axon development respectively, TIAM-1A was significantly more effective, suggesting that although different domains of TIAM-1 are required for regulating different sub-neuronal compartments, they perform better when they work together.

## TIAM-1 regulates polarized microtubule organization

Since polarized cytoskeleton organization and transport system are required for axonal and dendritic development [11], we examined whether TIAM-1 regulates the specific development of cytoskeleton structures to affect the polarized deployment of vesicular organelles such as synaptic vesicles and mitochondria in axonal and dendritic differentiation. We labeled

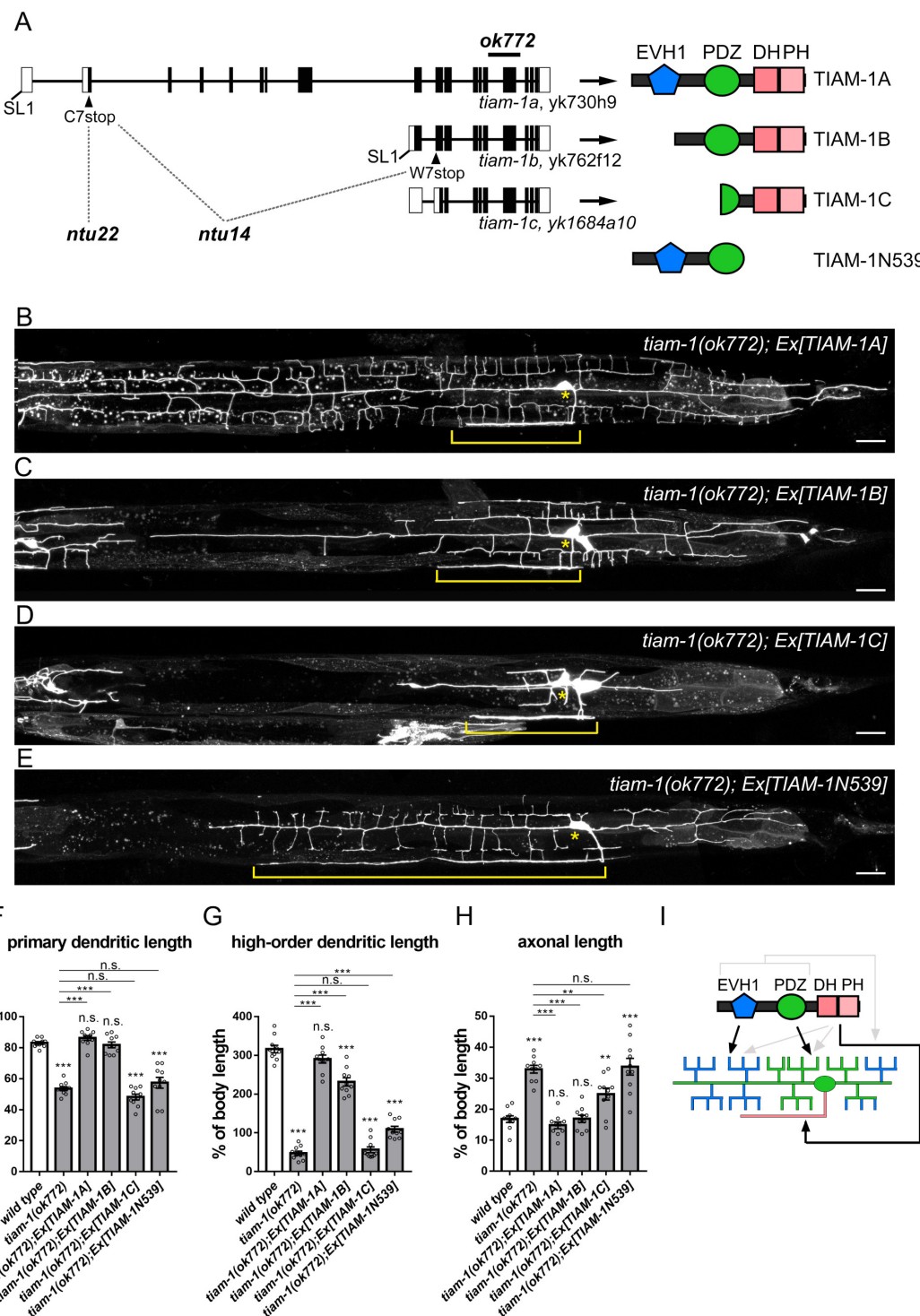

**Fig 2. Different TIAM-1 protein regions regulate the development of PVD subcompartments.** (A) Schematic diagram of three *tiam-1* cDNAs showing exons (black boxes), introns (lines), 5' and 3' untranslated regions (UTRs) (white boxes), SL1 trans-spliced leader sequence, and mutation sites of three *tiam-1* alleles (*ok772* and two CRISPR/Cas9 engineered alleles, *ntu14*, *and ntu22*). Functional domains of TIAM-1 isoforms are illustrated. (B-E) *tiam-1(ok772)* mutant PVD neuron with the expression of *Ex[Pdes-2::TIAM-1A]* (B), *Ex[Pdes-2::TIAM-1B]* (C), *Ex[Pdes-2::TIAM-1C]* (D), and *Ex[Pdes-2::TIAM-1N539]* (E) transgenes. PVD morphology is labeled by myr::mCherry (*ntuIs1*). Scale bar, 20 μm. (F-H) Quantification of the relative primary dendritic length (F), high-order dendritic length (G), and axonal length (H) in *tiam-1(ok772)* mutant expressing

different transgenes. One-way ANOVA. Error bar represents SEM. n = 10. n.s., not significant, $^*$P<0.05, $^{**}$P<0.01, $^{***}$P<0.001, compared to wild type or *tiam-1(ok772)*. (I) Proposed model for the function of different TIAM-1 domains in PVD neuron development: While the C-terminal DHPH domain is required for the axonal growth (the pink region), the middle PDZ domain is necessary for the growth of primary and proximal high-order dendrites (the green region). The N-terminal EVH1 domain is essential for the growth of distal high-order dendrites (the blue region). EVH1 and PDZ domains potentially enhance the C-terminal domain activity in suppressing axonal growth, and similarly the C-terminal domain potentially enhances the activity of EVH1 and PDZ domains in promoting dendritic development (grey arrows).

synaptic vesicles and mitochondria by RAB-3::GFP and TOMM-20::GFP, respectively [54,55]. We found a significant 1.7 times increase of synaptic vesicle clusters in the axon of *tiam-1 (ok772)* mutant compared to wild type (S3A, S3B, and S3E Fig). Meanwhile, the axonal mitochondrial number also increased more than two-fold in *tiam-1 (ok772)* when total mitochondrial number in all neurites remained the same as wild type (S3C–S3D and S3F–S3G Fig). Since the axon is relatively longer in *tiam-1* mutant, the density of synaptic vesicles and mitochondria in the axon showed no significant difference to wild type (S3H–S3I Fig).

Given that synaptic vesicles and mitochondria are cargos of kinesin motors, we examined the distribution of the plus-end motor UNC-116 (Kinesin-1) and UNC-104 (Kinesin-3), which are required for the transport of mitochondria and synaptic vesicles, respectively [24,56–58]. In PVD neurons, the anterior dendrite possesses MTs in minus-end-out orientation, while MTs are plus-end-out in the posterior dendrite and the axon [59]. These plus-end motors UNC-116 and UNC-104 localized to the axonal tips and the terminals of posterior dendritic branches of wild type (arrowheads in Figs 3A and S3J). In *tiam-1(ok772)*, their signals were severely diminished in the posterior dendrite but accumulated in the axon (Figs 3B–3C and S3J–S3L). These results indicated that the *tiam-1(ok772)* mutation disrupts the plus-end motor-mediated transport system in the dendrite but enhances it in the axon.

We next examined the microtubule polarity by tracing the movement of the microtubule plus-tip marker EBP-2::GFP. We found that the loss of TIAM-1 has no effect on the direction of EBP-2 comets in the anterior primary dendrite and axon, but the EBP-2 puncta running into the axon significantly increased, suggesting that there are more growing microtubules in the axon in *tiam-1(ok772)* mutant (Fig 3E–3H). Interestingly, in *tiam-1(ok772)* mutant, most of the EBP-2 comets ran in a reversed direction compared to wild type in the posterior dendrite (Fig 3D–3E, 3I and S1 and S2 Videos), suggesting that microtubule orientation is reversed in the posterior dendrite of *tiam-1(ok772)* mutant. Taken together, these results support the idea that TIAM-1 is required for organizing polarized microtubules to direct plus-end motors and their cargos.

## TIAM-1 colocalizes with MT and actin bundles

Proper MT organization requires interacting proteins to tether and stabilize MTs to the cell membrane or cortical actin filaments [60,61]. In order to understand how TIAM-1 regulate MT organization, we constructed and expressed a GFP-tagged full-length TIAM-1A to examine its subcellular localization. We found that TIAM-1A::GFP specifically localizes to the somatodendritic compartment with punctate patterns in the primary and high-order dendrites but does not localize to the axon (Fig 4A). We also labeled TIAM-1B and TIAM-1C as well as TIAM-1N539 with GFP to examine their localization in *tiam-1* mutant PVD neuron. Different from the punctate TIAM-1A::GFP, TIAM-1B::GFP was diffuse and mainly localized in primary dendrites, the cell body, and the axon (S4A and S4B Fig). TIAM-1C::GFP was enriched in the cell body with a small amount diffuse in neurites (S4C Fig). TIAM-1N539::GFP was punctate and specifically in dendrites similar to TIAM-1A::GFP, suggesting that the formation of dendritic-specific puncta largely depends on the EVH1 domain (S4D Fig).

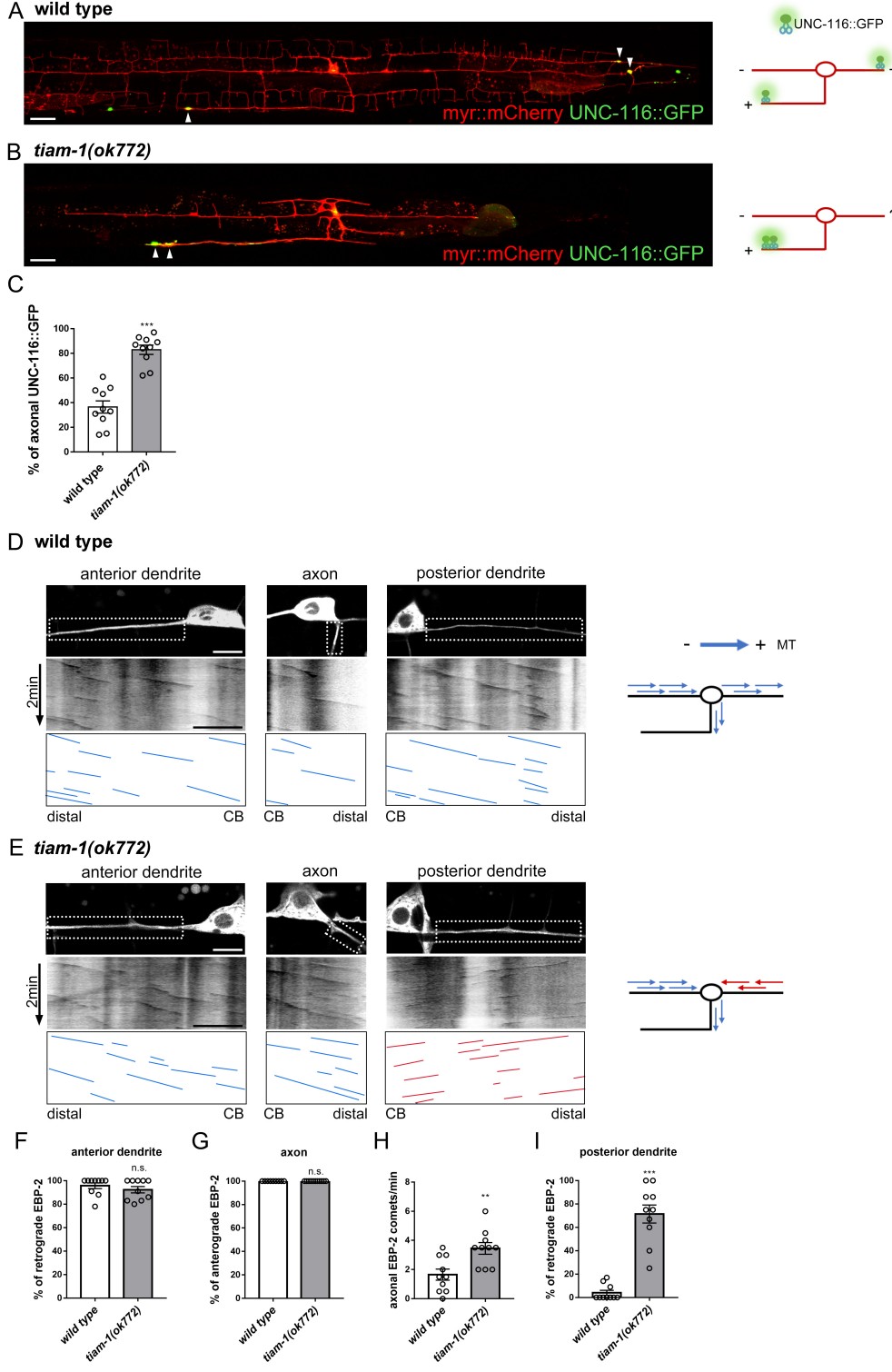

**Fig 3. TIAM-1 is required for developing the organized microtubules in PVD neuron.** (A and B) PVD neurons expressing UNC-116::GFP and myr::mCherry (*ntuIs2*) of wild type (A) and *tiam-1(ok772)* (B), with schematic diagrams of phenotypes. Arrowheads indicate UNC-116::GFP signal. Scale bar, 20 μm. (C) Quantification of axonal UNC-116::GFP intensity. Student's *t*-test. Error bar represents SEM. n = 10. n.s., not significant, ***P<0.001, compared to wild type. (D and E) Top: PVD neurons expressing the microtubule plus-end marker EBP-2::GFP (*hrtSi5*) of wild type (D) and *tiam-1(ok772)* (E). Bottom: The corresponding kymographs. Right: schematic diagrams

of phenotypes. CB, cell body. Scale bar, 5 μm. (F-I) Quantification of the percentage of retrograde EBP-2 growth events in anterior dendrites (F), the percentage of anterograde EBP-2 growth events in axons (G), the number of axonal EBP-2 growth events (H), and the percentage of retrograde EBP-2 growth events in posterior dendrites (I). Student's *t*-test. Error bar represents SEM. n = 10. n.s., not significant, *P<0.05, **P<0.01, ***P<0.001, compared to wild type.

We further examined the localization of TIAM-1A::GFP puncta with MT (labeled by RFP:: TBA-1) and actin filaments (labeled by RFP::UtrCH). Compared to GFP control, a much higher percentage of TIAM-1A::GFP is colocalized with either TBA-1 or UtrCH, suggesting that rather than exhibit random distribution, TIAM-1A::GFP has a better colocalization with MT and actin filaments (Fig 4B–4E). In high-order dendritic regions, TIAM-1A::GFP puncta localized in the MT-rich region basal to actin-rich branches (S5A and S5B Fig). To further examine the relationship between TIAM-1 and cytoskeleton, we genetically disrupted the actin structure by using *act-4/Actin* mutant allele *dz222*, which was reported to reduce dendritic arborization and UtrCH intensity in PVD neurons [46]. In *act-4(dz222)* mutants, while the cortical actin bundles were disrupted, we found decreased TIAM-1A::GFP in the cell cortex and aggregated TIAM-1A::GFP signals with remaining actin structures (S5C Fig). With 3D surface reconstruction of the cell surface and TIAM-1A puncta, the distance between TIAM-1 and the cell surface was significantly increased in *act-4(dz222)* mutants (S5D Fig). These results suggest that TIAM-1 localization requires appropriate actin organization.

A recent study has shown that UNC-119 is required for anchoring MT to the cell cortex [25]. We further examined the relative localization pattern of TIAM-1 and UNC-119. Interestingly, we found UNC-119 was highly colocalized with TIAM-1A::GFP in dendritic branches (Fig 4F and 4G). According to these observations, it is likely that TIAM-1 is associated with UNC-119.

## TIAM-1 is essential for the organization of actin filaments and MTs

In wild-type PVD neurons, MTs labeled by RFP::TBA-1 were enriched in primary dendrites and the axon with a distribution pattern not overlapping with actin filaments labeled by GFP:: UtrCH, which mainly distributed in high-order dendrites (Fig 5A). In *tiam-1* mutant, the distribution of MTs were reduced in primary dendrites and strongly enhanced in the axon (Fig 5B and 5C). Strikingly, actin filaments highly accumulated in the primary dendrites and the axon in *tiam-1* mutant (Figs 5B, 5D and S6A). These ectopic actin filaments abnormally intermingled with MTs in primary dendrites (Fig 5B'). In contrast, wild-type cortical actin filaments surround MTs in primary dendrites (Fig 5A'). In wild-type cell body, MTs form a regular organization between the nucleus and cortical actin filaments (Fig 5A"), but, in *tiam-1* mutant, MTs detached from the cortex with irregular turns and circles (Figs 5B" and S6B). Taken together, these results suggested that TIAM-1 is required for the proper distribution of actin filaments and the organization of the cytoskeletons in PVD neuron.

## TIAM-1 interacts with UNC-119 and regulates its distribution

Since UNC-119 is required for microtubule organization [25], we wonder whether TIAM-1 regulates UNC-119 localization. In *tiam-1* mutant, the distribution of UNC-119 significantly increased in axonal regions compared to wild type (Fig 6A–6C). As UNC-119, the axonal intensity of UNC-33 also increased in *tiam-1* mutant (Fig 6D–6F). While the depletion of TIAM-1 caused the increased axonal distribution of UNC-119, we next wondered whether overexpression of TIAM-1 also affects its localization pattern. Interestingly, overexpression of TIAM-1A::GFP with UNC-119::mCherry caused both UNC-119::mCherry and TIAM-1A:: GFP to form larger puncta in distal dendritic region (Fig 6G–6I), suggesting that TIAM-1 and

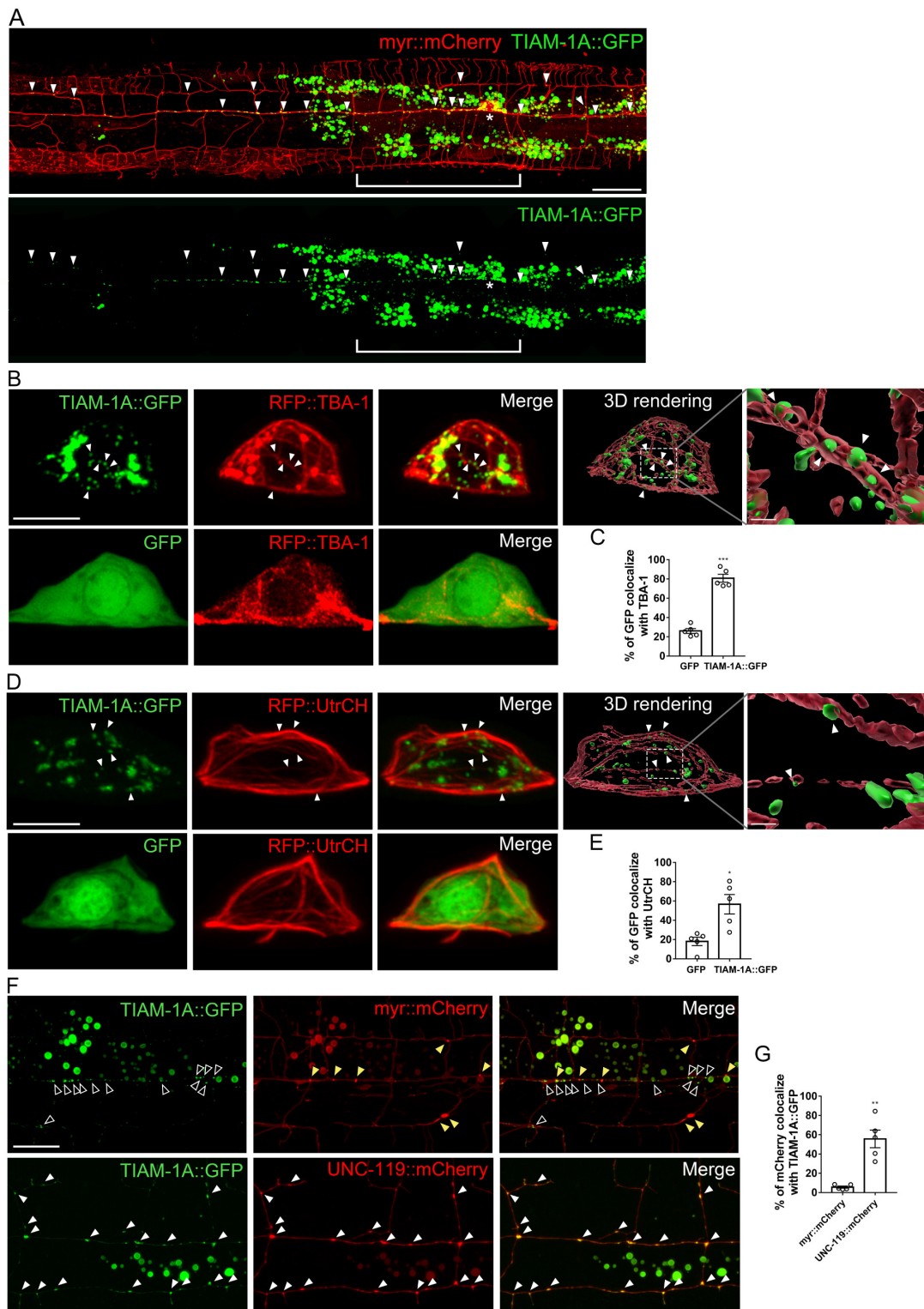

**Fig 4. TIAM-1 is localized to microtubules and actin filaments.** (A) Images of PVD neuron expressing TIAM-1A::GFP and myr::mCherry *(ntuEx278)*. TIAM-1A::GFP is specifically localized in dendrites (arrowheads), but not present in the axon (brackets). Scale bar, 20 μm. (B and D) PVD neuron cell body expressing the MT marker RFP::TBA-1 (B) or the F-actin marker RFP::UtrCH (D) in worms with TIAM-1A::GFP (top) or GFP control (bottom). Arrowheads indicate TIAM-1A::GFP signal. The 3D surface reconstructions are generated by Imaris software (top right). Scale bar, 5 μm or 0.3 μm (magnified images). (C

and E) Quantification of the percentage of GFP intensity colocalized with RFP::TBA-1 (C) or RFP::UtrCH (E). Student's *t*-test. Error bar represents SEM. n = 5. *P<0.05, ***P<0.001, compared to GFP. (F) PVD neurons expressing myr::mCherry (top) or UNC-119::mCherry (bottom) with TIAM-1A::GFP. Hollow arrowheads indicate the TIAM-1A::GFP signals, yellow arrowheads indicate myr::mCherry signals, and white arrowheads indicate the colocalization of TIAM-1A::GFP and UNC-119::mCherry signals. Scale bar, 10 μm. (G) Quantification of the percentage of mCherry colocalized with TIAM-1A::GFP signal. Student's *t*-test. Error bar represents SEM. n = 5. **P<0.01, compared to myr::mCherry.

UNC-119 have the potential to interact and regulate each other's distribution. We further performed bimolecular fluorescence complementation (BiFC) assay to examine the interaction by tagging Venus N-terminal fragment to UNC-119 and C-terminal fragment to TIAM-1A. Indeed, the BiFC signals were observed in PVD with a punctate pattern (Fig 6J and 6K). Since EVH1 and PDZ domains function as protein interaction domains [62,63], we expressed the N-terminal fragment TIAM-1N539 (tagged with HA) with UNC-119 (tagged with FLAG) in human embryonic kidney cells (HEK293T) to test co-immunoprecipitation. We found that TIAM-1N539 efficiently immunoprecipitated UNC-119 (Fig 6L). Together, these experiments suggested that TIAM-1 interacts with UNC-119 and regulates its distribution.

## TIAM-1 regulates axon development through UNC-119

Given that TIAM-1 colocalized and interacted with UNC-119, we hypothesized that TIAM-1 regulates axonal/dendritic polarized development by localizing UNC-119 in dendrites and preventing it from axon mislocalization. We tested if the elevation of the UNC-119 level is sufficient to promote axonal growth by expressing *unc-119* transgene and the results were positive (Fig 7A, 7B and 7G). We further genetically ablated *unc-119* in *tiam-1* mutant to test whether UNC-119 is required for the dysregulated axonal outgrowth by using the putative null mutant allele *ed3* [64]. In the *tiam-1; unc-119* double mutant, the ablation of *unc-119* effectively suppressed axonal length, compared to *tiam-1* single mutant (Fig 7C–7E and 7G). Regarding dendrite development, *tiam-1; unc-119* double mutation did not enhance the growth defects caused by *tiam-1* single mutation, suggesting they act in the same pathway in dendrite development (Fig 7C–7E and 7F). To sum up, these results showed that UNC-119 levels determine axon length and are required for TIAM-1 to regulate axonal growth.

## Discussion

Neuronal cytoskeletons are the basis for polarity establishment and the development of distinct axonal and dendritic structures to form functional synaptic connections. How cytoskeleton regulators promote polarized actin and MT organization in neuronal cells remains to be elucidated. Here, we examined the role of several actin regulators in axonal/dendritic development and identified UNC-60/Cofilin as a specific axonal regulator and WVE-1/WAVE as a specific dendritic regulator. Among these regulators, we find that the RacGEF TIAM-1 regulates both axonal and dendritic development but in opposite ways; it suppresses axonal growth but promotes dendritic formation. Our functional analysis shows that TIAM-1 directs distinct developmental patterns of neuronal subcompartments by its protein domains. While the N-terminal EVH1 domain of TIAM-1 promotes distal dendritic development, its middle PDZ domain is essential for primary dendrite formation. And, surprisingly, its C-terminal region is responsible for regulating axon development but not dendrite development. We further find TIAM-1 interacts with the cortical anchor complex component UNC-119 and colocalizes with actin filaments and MTs. Based on our findings, we propose a working model that TIAM-1 regulates MT and actin organization to direct axonal and dendritic development (Fig 7H). UNC-119 is essential for TIAM-1 to regulate axonal outgrowth.

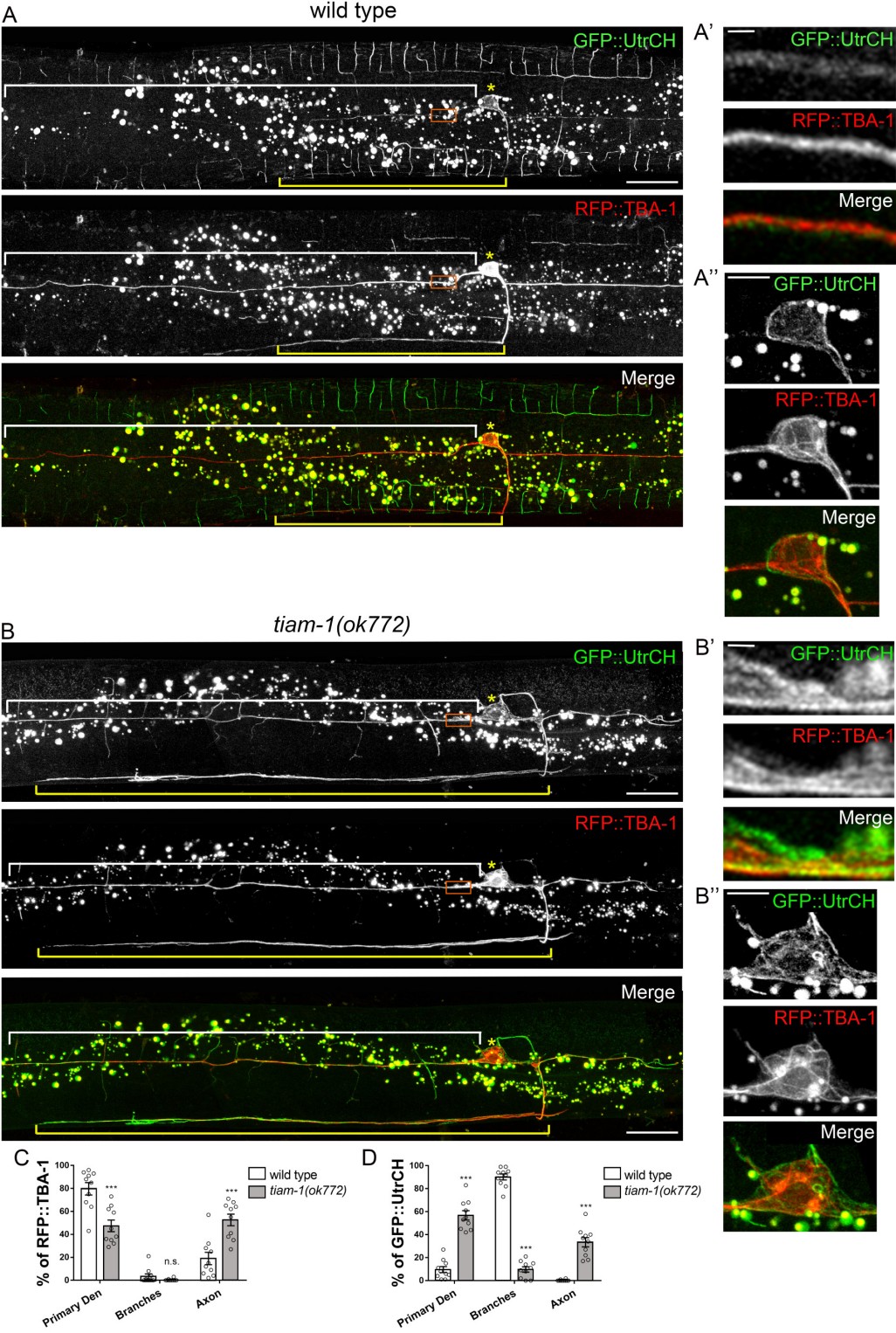

**Fig 5. TIAM-1 is required for the proper organization of MTs and actin filaments.** (A and B) PVD neurons expressing F-actin marker GFP::UtrCH and MT marker RFP::TBA-1 in wild type (A) and *tiam-1(ok772)* (B). In wild-type PVD, F-actin was mainly distributed in high-order dendritic branches, while MT was mainly distributed in the primary dendrites (white brackets) and the axon (yellow brackets). In *tiam-1(ok772)*, F-actin and MT were abnormally distributed in the primary

dendrite and aberrantly accumulated in the axon. Yellow asterisks indicate the cell body. The red box regions in (A) and (B) are magnified in (A') and (B'). The cell body in (A) and (B) are magnified in (A") and (B"). Notably, F-actin abnormally intermingled with MT in (B') and (B") compared to normal localization in cell cortex of wild type in (A') and (A"). Scale bar, 20 μm in A and B, 1 μm in A' and B', 5 μm in A" and B". (C and D) Percentage of GFP::UtrCH (C) and RFP::TBA-1 intensity (D) in different neuronal compartment. Error bar represents SEM. n = 10. n.s., not significant, *P<0.05, **P<0.01, ***P<0.001, Student's t-test, compared to wild type.

## Different TIAM-1 protein domains direct distinct axonal and dendritic development

In previous studies, TIAM-1 has been shown to regulate the development of dendrite and dendritic spine by its GEF activity in mice hippocampal neurons and worm PVD neurons [45,65]. But, Bülow group recently shows TIAM-1 GEF activity is not required in dendritic morphogenesis since a TIAM-1 mutation (T548F) that abolishes its GEF activity has no effect on dendrite development although the role of GEF domain in neuronal development remains unclear [46]. In this work, we surprisingly find the GEF domain-containing C-terminal fragment TIAM-1C regulates axon growth but not dendrite development. We further find the N-terminal fragment TIAM-1N539 without the GEF domain can restore a portion of dendrite development but not able to regulate axon development. These results suggest that the C-terminal GEF domain controls axon development, while EVH1 and PDZ domains regulate dendrite development. Interestingly, our results further indicate that EVH1 domain directs the development of distal high-order dendritic branches, and PDZ domain supports primary dendrite growth, respectively. Therefore, here we report that TIAM-1 is a multifunctional protein that directs distinct developmental patterns of neuronal subcompartments through its diverse functional domains.

## *tiam-1* isoforms play different roles in neuronal morphogenesis

In this study, we find that *tiam-1* isoforms have different physiological roles in neuronal morphogenesis. Similarly, there are two isoforms of *tiam-1* mammalian homolog *Tiam2*, *Tiam2l* (full-length form) and *Tiam2s* (short form), in mice and humans, functioning differently [66,67]. While *Tiam2l* suppresses *t*-haploid transmission, *Tiam2s* expression strongly enhances the transmission ratio of *t*-allele, suggesting that *tiam-1* isoforms act differently in spermatogenesis [66]. Moreover, the mammalian *Tiam1* has four isoforms that are detected in the cerebral cortex and hippocampus and their expressions are increased in Down syndrome mouse model, while the function of individual isoforms in the nervous system is unknown [68]. In neuronal development, we show that the full-length *tiam-1a* is responsible for high-order dendrite formation, while *tiam-1b* is required in primary dendrite formation. Since TIAM-1B lacks the EVH1 domain, which is often present in actin cytoskeleton regulators like Ena/VASP and WASP [62], it is possible that TIAM-1A utilizes this region to promote actin organization in distal dendrites. Actin cytoskeletons appear to interact with MT bundles differently in primary dendrites, where actin filaments surround MT bundles under the cell membrane, in comparison with actin-rich distal dendritic processes emanated from MTs (Fig 5A). We propose that the loss of the EVH1 domain makes TIAM-1B relocate to primary dendritic region and change its way to interact with cytoskeletons and promote primary dendrite formation (S4B Fig).

## TIAM-1 organizes actin and MT cytoskeletons for axonal/dendritic polarized development

Although the MT cytoskeleton supports neuronal morphogenesis and serves as tracks for directional trafficking of cargos to promote dendritic and axonal differentiation, the molecular

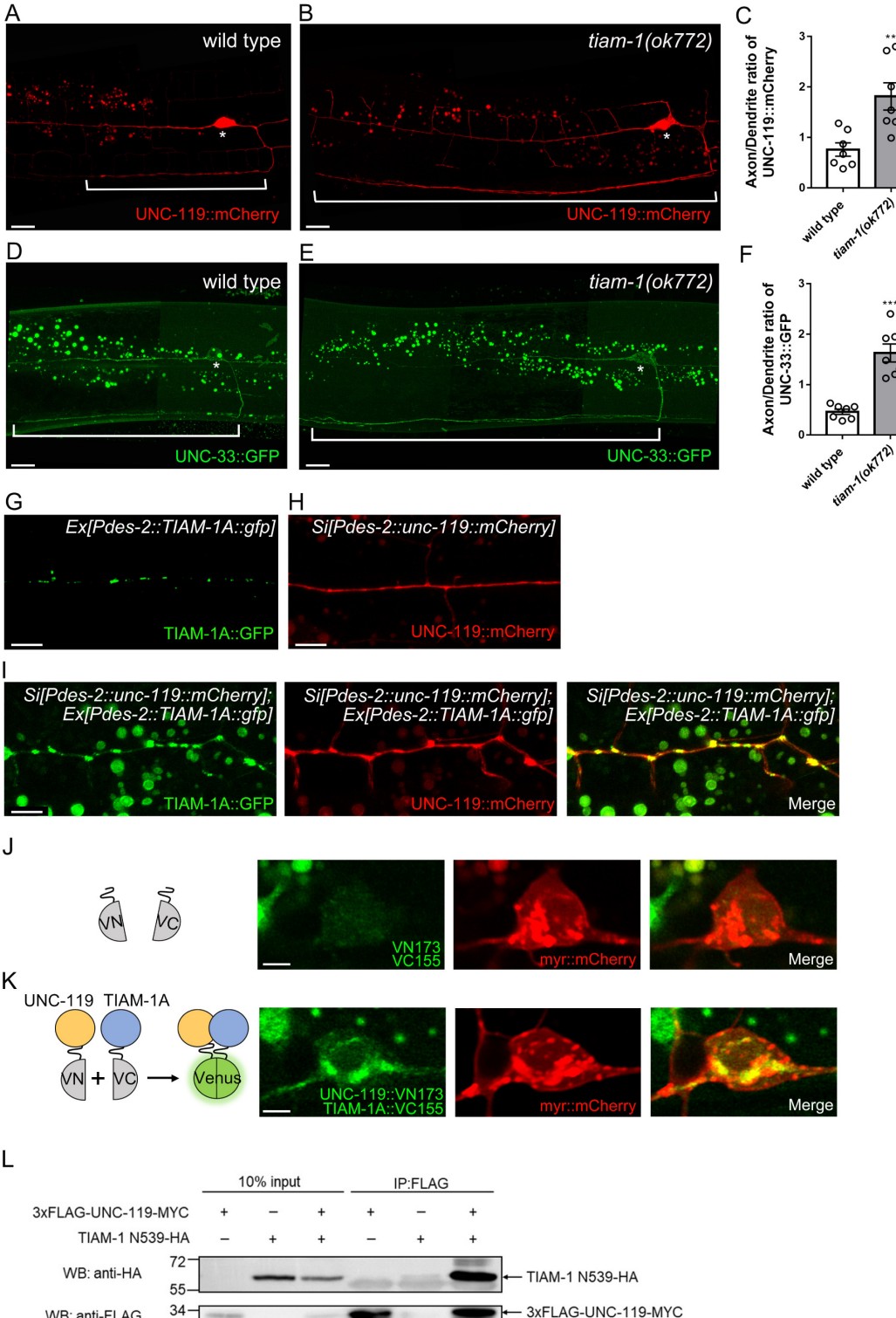

**Fig 6. TIAM-1 interacts with UNC-119 and regulates the distribution of UNC-119 in PVD neurons.** (A and B) UNC-119::
mCherry expressed by a single-copy transgene in PVD neuron in wild type (A) and *tiam-1(ok772)* (B). Brackets indicate the axon
and asterisks indicate the cell body. Scale bar, 10 μm. (C) The ratio of axonal to dendritic UNC-119::mCherry intensity. Error bar,
SEM, n = 7. **P<0.01, Student's *t*-test, compared to wild type. (D and E) UNC-33::GFP expressed by a single-copy transgene in

PVD neuron in wild type (D) and *tiam-1(ok772)* (E). Brackets indicate axon and asterisks indicate cell body. Scale bar, 10 μm. (F) The ratio of axonal to dendritic UNC-33::GFP intensity. Error bar, SEM, n = 7. ***P<0.001, Student's *t*-test, compared to wild type. (G and H) Image of PVD neuron expressing TIAM-1A::GFP alone (G) or UNC-119::mCherry alone (*ntuSi9*) (H). Scale bar, 5 μm. (I) Images of PVD neuron coexpressing TIAM-1A::GFP with UNC-119::mCherry (*ntuSi9*). Overexpressing TIAM-1A::GFP causes both UNC-119::mCherry and TIAM-1A::GFP to form large puncta. Scale bar, 5 μm. (J and K) PVD neuron expressing control Venus fragments (J) or UNC-119::VN173 and TIAM-1A::VN155 (K) with myr::mCherry (*ntuIs13*). Left schematic represents combinations of expressed Venus fragments in bimolecular fluorescence complementation (BiFC) assays. VN173, Venus N-terminal 1–173 residues. VN155, Venus C-terminal 155–238 residues. Scale bar, 2 μm. (L) Western blot of co-immunoprecipitation (IP) experiments. Asterisks marked the presumably partially degraded form of FLAG-UNC-119-MYC.

mechanisms that regulate distinct axonal/dendritic MT organizations remain largely unknown [10,15]. In neuronal cells, different actin structures interact and coordinate with MTs through crosslinking proteins and complexes [1]. It is likely that some crosslinking mechanisms regulate MT organization and coordinate dynamics of actin structures for axon or dendrite development and positioning organelles. Indeed, UNC-119 interacts with UNC-33 and UNC-44 to form a complex that anchors MTs to cell cortex [25]. Also, we recently showed that UNC-33 assembles a high-order MT organization for axonal transport [24].

In this work, we show that TIAM-1 is essential to maintain plus-end-out MTs in posterior dendrites and regulates axon specific cargos including synaptic vesicles and mitochondria. Notably, we find that TIAM-1 puncta localize with MTs and actin bundles and also interact with UNC-119. Based on these findings, it is possible that TIAM-1 stabilizes dendritic plus-end-out MTs through the cortical anchor complex in the posterior dendrite. Consistent to this model, dendritic microtubule orientation is indeed disrupted in cortical anchor complex mutants, including *unc-33*, *unc-44*, and *unc-119* [25,59]. In addition, since the minus-end binding protein PTRN-1/CAMSAP proteins stabilize MTs and regulate neuronal polarity [69–73], it is interesting to investigate the potential regulation between TIAM-1 and PTRN-1 in future studies.

### TIAM-1 regulates axon development via UNC-119

We find the distribution of cortical anchor complex subunits UNC-119 and UNC-33 shifts to the axon by ablating *tiam-1* (Fig 6A–6F). UNC-119 has been reported to be enriched in nervous system in invertebrates and vertebrates [64,74] and participate in axonal development [74,75]. Here we show that the loss of UNC-119 leads to shorter axons while overexpressing UNC-119 induces the extension of the axon, suggesting that UNC-119 is necessary and sufficient to promote axonal growth. It is possible that the increase of axonal UNC-119 recruits cortical anchor complex and stabilizes the tight interaction between cortical actin rings and MT arrays in axons. Consequently, the more MTs are stabilized as tracks, the more motors and cargos are recruited into the axons, and thus it facilitates axonal growth. Since TIAM-1 specifically distributes in dendrites and strongly interacts with UNC-119, the interaction between TIAM-1 and UNC-119 may stabilize the cortical anchor complex in dendrite and maintain the balance between dendritic and axonal development.

## Materials and methods

### Strains

Worms were raised on NGM plates seeded with *Escherichia coli* OP50 at 20˚C. The Bristol N2 strain was used as wild type and all transgenic strains created for this study are shown in **S1 Table**.

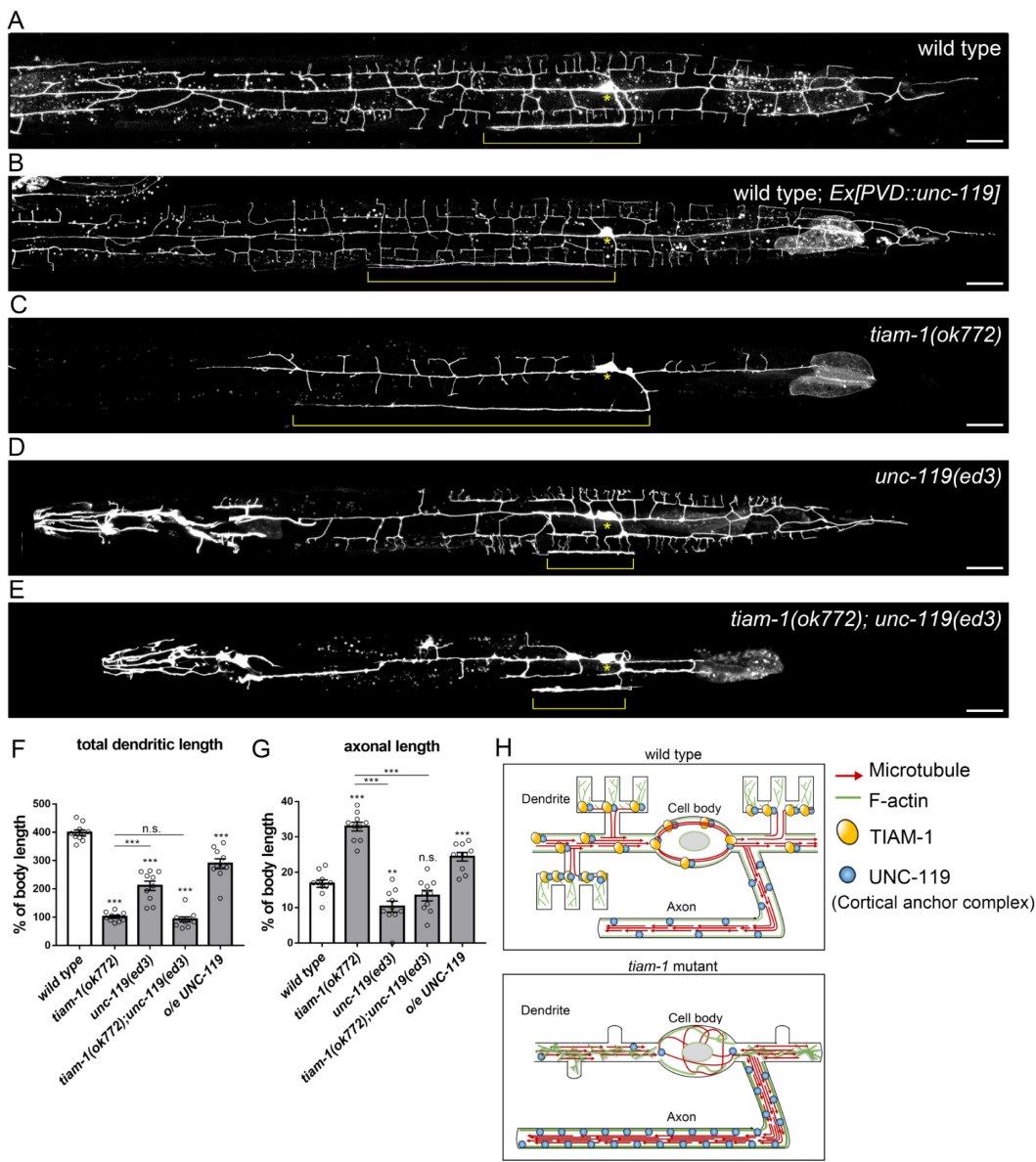

**Fig 7. TIAM-1 regulates the development of axon through UNC-119.** (A-E) PVD neuron expressing myr::mCherry under *Pdes-2* promoter (*ntuIs1*) of wild type (A), *Ex[Pdes-2::unc-119]* (B), *tiam-1(ok772) (C)*, *unc-119(ed3) (D)*, and *tiam-1(ok772); unc-119(ed3)* (E). Brackets indicate the axons and asterisks indicate the cell body. Scale bar, 20 μm. (F and G) Quantification of the relative total dendritic length (F) and relative axonal length (G) in the indicated genotypes. One-way ANOVA, Error bar, SEM. n.s., not significant, *P<0.05, **P<0.01, ***P<0.001, compared to wild type or *tiam-1(ok772)*. n = 10. (H) A proposed model of TIAM-1 in regulating microtubule organization. In wild type, TIAM-1 localizes UNC-119 to stabilize the organization of the cytoskeleton in somatodendritic regions. In *tiam-1* mutant, MTs and actin filaments are disorganized in dendrites and the cell body, while mislocalized UNC-119 accumulates with MTs and actin filaments in the axon.

## Constructs and transgenic worms

Expression clones were made in the pSM vector, a derivative of pPD49.26 (A. Fire) with extra cloning sites (*S. McCarroll* and *C. I. Bargmann*, personal communication). The *Pdes-2* promoter was used for PVD-specific expression, and *Pdes-2* promoter fragment was amplified as previously described with *Sph*I and *Asc*I sites [76] (Phusion, NEB). Details for plasmids that used in this study to generate transgenic worms can be found in **S1 Table**.

For *tiam-1* rescue experiment, *TIAM-1A*, *TIAM-1B*, *TIAM-1C*, or *TIAM-1N539* were constructed with *Pdes-2* as pPW13, pCL16, pCL17, and pCL14 which were injected separately into strain OCY580: *tiam-1(ok772); ntuIs1* to generate strains OCY1141: *tiam-1(ok772); ntuIs1; Ex [Pdes2*::TIAM-1A], OCY875: *tiam-1(ok772); ntuIs1; Ex[Pdes2*::TIAM-1B], OCY1094: *tiam-1 (ok772); ntuIs1; Ex[Pdes2*::TIAM-1C], and OCY1320: *tiam-1(ok772); ntuIs1; Ex[Pdes2*::TIAM-1N539].

To perform *tiam-1* rescue experiment by single-copy transgene insertion, the single copy of *Pdes-2*::TIAM-1A::gfp, *Pdes-2*::TIAM-1B::gfp, *Pdes-2*::TIAM-1C::gfp, and *Pdes-2*::TIAM-1N539:: gfp were separately integrated into the genome by phiC31-mediated recombination in *tiam-1 (ok772)*, *ntuIs13* (OCY2231) in BRC0566 as described [53] to generate *ntuSi14* (OCY2202), *ntuSi15* (OCY2232), *ntuSi16* (OCY2233), and *ntuSi17* (OCY2234).

To visualize the localization of TIAM-1 in PVD, *Pdes-2*::TIAM-1A::gfp (pPW12) was injected into *ntuIs13* worms to generate OCY2130. To visualize PVD neuronal morphology, we generated *ntuIs13* by integrating extrachromosomal array containing *Pdes-2*::myr::mCherry (pHW5) [24] into N2 genome by UV irradiation at 100 μJ/cm$^2$ [77].

To visualize the localization of TIAM-1::GFP with MTs or F-actin, a single copy of *Pdes-2*:: *TIAM-1A*::gfp was integrated into the genome in BRC0566 as described [53] to generate *ntuSi1* (OCY1313). *Pser-2prom3*::tagRFP::tba-1 and *Pser2prom3*::tagRFP::UtrCH (kind gifts of Hannes E Bülow lab) were injected separately into OCY1313 to generate OCY1538 and OCY2132. To compare TIAM-1A::GFP with GFP, the *Pdes-2*::gfp was co-injected with *Pser2-prom3*::tagRFP::tba-1 and *Pser2prom3*::tagRFP::UtrCH into N2 to generate OCY1503 and OCY1504, respectively.

To co-label UNC-119 and TIAM-1, UNC-119 was amplified by PCR from cDNA library and cloned under *Pdes-2* promoter. *Pdes-2*::unc-119::mCherry (pCL40) was injected into OCY1313 to generate OCY1630.

To examine how TIAM-1 overexpression affects the pattern of UNC-119, *Pdes-2*::TIAM-1A::gfp (pPW12) was injected into OCY1629 to generate OCY2131. OCY1629 was generated by integrating a single copy of *Pdes-2*::unc-119::mCherry as described [53].

To co-label F-actin and MT in PVD neurons, *Pdes-2*::gfp::UtrCH (pCL62) and *Pser2prom3*:: tagRFP::tba-1 (courtesy of Hannes E Bülow lab) were co-injected into N2 worms to generate OCY2034.

To examine how UNC-119 overexpression influences PVD neuronal development, *Pdes-2*:: unc-119 (pCL55) was injected into OCY295: *ntuIs1* to generate OCY1675.

To perform BiFC assay, VN173 (Venus N-terminal 1–173 residues) and VC155 (Venus C-terminal 155–238 residues, A206K) were respectively amplified from pAH120 and pAH121, which are gifts from Prof. Ao-Lin Hsu. *Pdes-2*::VN173 (pCL68) and *Pdes-2*::VC155 (pCL67) were co-injected into *ntuIs13* (as a negative control) to generate OCY2134. *Pdes-2*::unc-119:: VN173 and *Pdes-2*::TIAM-1A::VN155 were co-injected into *ntuIs13* to generate OCY2135.

## CRISPR/Cas9 mediated genome editing

For introducing nonsense mutations into endogenous C7 loci of *tiam-1a(ntu22)*, the CRISPR/ Cas9 mediated genome editing method will be utilized [78]. Briefly, Cas9 protein (10 μg/μl), tracrRNA (4 μg/μl), crRNA_*rol-6* (4 μg/μl), crRNA_ *tiam-1a* (4 μg/μl) were mixed and incubated for 10 mins at 37°C. Then the mixture was added with the repair DNA templates (1 μg/ μl) and injected into *ntuIs1* (OCY295 strain). Rollers from the F1 generation were selected. F2 worms were lysed and the putative mutation sites in *tiam-1* locus were sequenced. To simultaneously edit C7 codon of *tiam-1a* and W7 codon of *tiam-1b* into stop codons to generate *tiam-1(ntu14)*, crRNA_ *tiam-1b* and the repair DNA templates for *tiam-1b* were used. All

reagents were ordered from Integrated DNA Technologies (IDT). The crRNA sequences and repair oligo sequences are shown in **S1 Table.**

## Immunoprecipitation and immunoblotting

To express TIAM-1 N-terminal domain and UNC-119 for Co-IP assays, Human Embryonic Kidney Cells 293 (HEK293T) were cultured with a standard protocol in DMEM supplemented with 10% FBS. $2 \times 10^5$ cells/well (in a 6-well dish) were transfected with the expression plasmid containing 1.5 μg TIAM-1N539 and 1 μg 3xFLAG-UNC-119-MYC using HyFectin transfection reagent kit. Two days after transfection, cells were lysed in 250 μl of lysis buffer (50mM Tris-Base, 150mM NaCl, 1% IGEPAL CA-630, pH 7.4, protease inhibitor cocktail (MedChem-Express)), and the lysates were then sonicated for 10 minutes for fully lysis. The resulting lysates were centrifuged at 16,200 ×g for 10 minutes at 4˚C, and the supernatants were incubated with 20 μl of anti-FLAG M2 Affinity Gel (Sigma-Aldrich) at 4˚C with agitation overnight. After washing three times with the lysis buffer, the samples were eluted with SDS loading buffer and were boiled for 3 min. SDS-PAGE and western blot were then performed using standard protocols. Anti-HA (Proteintech) and anti-FLAG M2 (Sigma-Aldrich) mouse antibodies were used at 1:2000 dilutions and 1:1000 dilutions, respectively, and HRP-conjugated goat antibodies to mouse were used at 1:10,000 dilutions (Proteintech). Details for plasmids used in co-IP assays can be found in **S1 Table**.

## Image acquisition

For quantification, fluorescence images of live *C. elegans* were acquired using Zeiss Axio Imager microscope (Carl Zeiss, Germany) with X-cite light source and 63x/1.4NA oil objective and Evolve 512 EMCCD camera (photometrics) at 23˚C. The z stacks were collected and maximum intensity projections were used for further analysis. For each worm, 4–5 images were stitched together to cover the whole body (Adobe Photoshop). Worm images were straightened and measured by ImageJ. Worms at mid-L4 stage were immobilized with 2–3 μl of 5 mM levamisole (Sigma-Aldrich) on 3% agarose pads.

For confocal images in Figs 1–3 and 7 and S1–S3, worms were immobilized with 25mM levamisole on 8% agarose pads and imaged by a Zeiss LSM710 microscope (Carl Zeiss, Germany) with 63x/1.4NA oil objective with 488 nm and 561 nm laser lines.

For confocal images of Airyscan mode in Figs 4–6 and S4 and S5, worms were immobilized with 50mM levamisole on 8% agarose pads and imaged by Zeiss LSM880 confocal microscope (Airyscan SR mode).

Confocal images were acquired as Z-stack serials, and several images were taken and stitched together to cover the PVD neuron. Images were processed by Zen software to project maximum intensity projections for figures and further quantification.

## Quantification and statistical analysis

**Neuronal morphology analysis.**   For relative axonal or dendritic length, images of 10 worms for each genotype at mid-L4 stage were scored for their total dendritic length and axonal length as well as body length (from mouth to anus) by ImageJ. The dendritic or axonal length was divided by body length to get the relative dendritic or axonal length (Figs 1, 2, 7, S1 and S2).

**Quantification of mitochondria and synaptic vesicles.**   For the distribution of mitochondria and the number of synaptic vesicles, mitochondria and synaptic vesicle clusters in PVDs of 10 worms for each genotype at mid-L4 stage were scored. Numbers of synaptic vesicle clusters, total mitochondrial numbers in all neurites, percentages of axonal mitochondrial (axonal

mitochondrial number/ total neurite mitochondrial number), the density of synaptic vesicle (synaptic vesicle number/ axonal length), and density of mitochondria in axon (mitochondrial number/ axonal length) were quantified (S3 Fig).

**Quantification of motor fluorescence signals.** For quantification of UNC-116::GFP and UNC-104::GFP, the GFP signals in the posterior dendrites and axons of 10 worms were circled as regions of interest (ROIs) by ImageJ to acquire total fluorenscence. Additional ROIs with the same size were measured as background autofluorescence inside the worm. Axonal and dendritic GFP intensity values were acquired by subtracting the background fluorescence from total fluorescence. The percentages of axonal UNC-116::GFP or UNC-104::GFP were calculated by the formula: axonal GFP intensity/ (dendritic + axonal GFP intensity) ×100% (Figs 3 and S3).

**Quantification of colocalization.** For the colocalization analysis between TIAM-1A::GFP (or GFP) and RFP::TBA-1 (or RFP::UtrCH), 16–18 z-stack Airyscan confocal images of the cell body region (~6 μM in z axis) were analyzed by Imaris 9.9.0. The intensity threshold that best represents the signal level for both GFP and RFP channels were set in the "Coloc" tool, and the percentages of GFP intensity of voxels colocalized with RFP were calculated (Fig 4B-4E).

For the colocalization analysis between TIAM-1A::GFP and UNC-119::mCherry (or myr:: mCherry), 20–22 z-stack images of dendrites (~7 μM in z axis) were analyzed. 3D regions of dendritic branches were defined by Imaris "surface" tool to analyze TIAM-1A::GFP and UNC-119::mCherry in these regions. Percentages of mCherry intensity of voxels with GFP were calculated (Fig 4F-4G).

**Quantification of cytoskeletons.** For quantification of RFP::TBA-1 and GFP::UtrCH, the intensity of both fluorescence reporters was quantified by using ImageJ. The segmented line tool was used to trace RFP::TBA-1 and GFP::UtrCH signals in the whole neuron including primary dendrites, high-ordered branches, and axons to create ROIs and the line width is 10 pixels (= 1.6 μm). To measure background from worm body, additional ROIs of the equal size areas were measured as background autofluorescence in the neighboring region inside the worm avoiding neurites. The RFP::TBA-1 and GFP::UtrCH signals in three neuronal compartments (primary dendrite, high-ordered branches, and axon) were acquired by subtracting background intensity. Ratios of RFP::TBA-1 and GFP::UtrCH in three compartments were calculated (signal in a sub-neuronal compartment was divided by the sum of three neuronal compartments) and analyzed (Fig 5C and 5D).

**Quantification of UNC-119 and UNC-33.** For the quantification of UNC-119::mCherry and UNC-33::GFP, the intensity of both fluorescence reporters was quantified using ZEN3.4 (blue edition). The Draw Spline Contour tool was used to draw outlines of neurites to create ROIs. The signals in the axon and the 60 μm of the anterior primary dendrite (start from the cell body) were measured. To measure background levels, additional ROIs of equal areas were defined in a neighboring region avoiding neurites inside the worm. The UNC-119::mCherry and UNC-33::GFP intensity were subtracted by background signals and divided by the area of ROI to calculate average intensity. The average intensity of axonal UNC-119::mCherry and UNC-33::GFP were divided by dendritic signals to generate the axon/dendrite ratio of UNC-119::mCherry and UNC-33::GFP (Fig 6).

**Quantification of EBP-2 comet direction.** EBP-2::GFP was recorded at the rate (1 frame per second) for 2 minutes to further generate Kymographs by ImageJ with the "Multi Kymograph" plugin. To quantify EBP-2::GFP comet direction, the proximal primary dendrite region within 50 μm from cell body was recorded. For scoring axonal EBP-2::GFP comet events, the comets that ran into the axon within 15 μm from cell body were counted (Fig 3).

**Analysis of the distance from the TIAM-1A::GFP to the cell surface.** To analyze the distance between TIAM-1A::GFP puncta and the cell surface, the 3D surface reconstruction of

TIAM-1A::GFP and the cell surface were built by Imaris 9.9.0. The 'Surfaces' function was used to build the TIAM-1 puncta and the cell surface according to the TIAM-1A::GFP and RFP::UtrCH signals, respectively. After building the 3D surface reconstruction, the 'short distance' in Statistics tools was used and the distance of each TIAM-1 puncta to cell surface was quantified to generate S5 Fig.

One-way ANOVA, Student's *t*-test, or Fisher's exact test were performed for statistics. Diagrams were made by GraphPad Prism 7.0 software.

## Supporting information

**S1 Fig. Actin regulators that cause no influence or mild defects in PVD neuron development.** (A-C) PVD neurons expressing myr::mCherry (*ntuIs1*) in *fhod-1(tm2363)* (A), *wsp-1 (gm324)* (B), and *pfn-1(ok808)* (C). Brackets indicate the axons and asterisks indicate the cell body. Scale bar, 20 μm. (D and E) Quantification of total dendritic length (D) and axonal length (E) relative to body length in the indicated genotypes. One-way ANOVA, Error bar represents SEM. n = 10. n.s., not significant, *P<0.05, compared to wild type. (EPS)

**S2 Fig. TIAM-1 isoforms have different roles in neuronal development.** (A-B) PVD neurons expressing myr::mCherry (*ntuIs1*) in *tiam-1(ntu22)* (A) and *tiam-1(ntu14)* (B). Scale bar, 20 μm. (C-F) *tiam-1(ok772)* mutant PVD neurons with the expression of different single-copy transgene, including *Si[Pdes-2::TIAM-1A::gfp]* (C), *Si[Pdes-2::TIAM-1B::gfp]* (D), *Si[Pdes-2:: TIAM-1C::gfp]* (E), and *Si[Pdes-2::TIAM-1N539::gfp]* (F). PVD morphology is labeled by myr:: mCherry (*ntuIs13*). Scale bar, 20 μm. Brackets indicate the axons and asterisks indicate the cell body. (G-I) Quantification of the relative primary dendritic length (G), high-order dendritic length (H), and axonal length (I) in the indicated genotypes. One-way ANOVA. Error bar represents SEM. n = 10. n.s., not significant, *P<0.05, **P<0.01, ***P<0.001, the left side of dash lines are compared to wild type (*ntuIs13*) or *tiam-1(ok772), ntuIs13* and the right side of dash lines are compared to wild type (*ntuIs1*) or *tiam-1(ok772); ntuIs1*. (EPS)

**S3 Fig. TIAM-1 is required for the proper distribution of synaptic vesicles and mitochondria as well as UNC-104/Kinesin-3.** (A and B) Images of a PVD neuron expressing synaptic vesicle marker RAB-3::GFP and myr::mCherry (*wyEx5216*) in wild type (A) and *tiam-1 (ok772)* (B). Arrowheads indicate synaptic vesicle clusters. Scale bar, 20 μm. (C and D) Images of a PVD neuron expressing mitochondria marker TOMM-20::GFP and myr::mCherry (*ntuIs1*) in wild type (C) and *tiam-1(ok772)* (D). Arrowheads indicate mitochondria. Scale bar, 20 μm. (E-I) Quantification of synaptic vesicle cluster number (E), the percentage of axonal mitochondria (F), the number of mitochondria in all neurites (G), the density of synaptic vesicle in axon (H), and the density of axonal mitochondria (I). Error bar represents SEM, n = 10. Student's *t*-test, n.s., not significant, ***P<0.001, compared to wild type. (J and K) PVD neurons expressing UNC-104::GFP and myr::mCherry (*ntuEx138*) of wild type (J) and *tiam-1 (ok772)* (K). Arrowheads indicate UNC-104::GFP signals. Scale bar, 20 μm. (L) Percentage of axonal UNC-104::GFP intensity. Error bar represents SEM, n = 10. Student's *t*-test, ***P<0.001, compared to wild type. (EPS)

**S4 Fig. TIAM-1 isoforms show different expression patterns in PVD neurons.** (A-D) Images of a *tiam-1(ok772)* mutant PVD neuron expressing *Si[TIAM-1A::GFP]* (A), *Si[TIAM-1B::GFP]* (B), *Si[TIAM-1C::GFP]* (C), *Si[TIAM-1N539::GFP]* (D). PVD morphology is labeled by myr::mCherry (*ntuIs13*). Arrowheads indicate TIAM-1A::GFP or TIAM-1N539::GFP

puncta, brackets indicate the axons and asterisks indicate the cell body. Scale bar, 20 μm.
(EPS)

**S5 Fig. The distribution of TIAM-1A::GFP is affected by the organization of the actin fila-ment.** (A and B) PVD neuron high-order dendrites expressing the MT marker RFP::TBA-1 (A) or the F-actin marker RFP::UtrCH (B) in worms with TIAM-1A::GFP. Arrowheads indicate TIAM-1A::GFP signal. Scale bar, 5 μm. (C) PVD neuron cell bodies expressing F-actin marker RFP::UtrCH and TIAM-1A::GFP in wild type and *act-4 (dz222)* (left) and the corresponding 3D reconstructions of TIAM-1A::GFP and cell surface (right). Top: the top view of the cell body. Bottom: the side view of the cell body. Scale bar, 2 μm. The magnified image represents the shortest distance between TIAM-1A::GFP puncta and the cell surface (the white vertical line), Scale bar, 0.2 μm. (D) Quantification shows the shortest distance between TIAM-1A::GFP puncta and the cell surface in wild type and *act-4 (dz222)*. Error bar represents SEM, n = 189 (wild type) and n = 144 (*dz222*), Student's *t*-test, ***P<0.001, compared to wild type.
(EPS)

**S6 Fig. TIAM-1 is required for cytoskeleton organization in PVD neuron.** (A and B) Penetrance of actin filament mislocalization in primary dendrites (A) and penetrance of disorganized cytoskeletons in the cell body (B). n = 82 (wild type) and n = 61 (*ok772*). Fisher's exact test, ***P<0.001, compared to wild type.
(EPS)

**S1 Table. List of strains, plasmids, and oligonucleotides used in this study.**
(DOCX)

**S2 Table. Numerical data for figures in this study.**
(XLSX)

**S1 Video. EBP-2::GFP comets in wild type PVD neuron.**
(MP4)

**S2 Video. EBP-2::GFP comets in *tiam-1(ok772)* PVD neuron.**
(MP4)

## Acknowledgments

We thank imaging core facility of the First Core Labs, and Helene Minyi Liu, National Taiwan University and Hsun Li, Chien lab, Academia Sinica, Bülow lab, Albert Einstein College of Medicine, Yuji Kohara, National Institute of Genetics, and Ao-Lin Hsu, National Yang-Ming Chiao-Tung University for technical support and reagents. We thank Cheng-Ting Chien, Academia Sinica, and Jing-Jer Lin, National Taiwan University for discussion on the manuscript. We thank John Wang, Academia Sinica for phiC31 integration technique. We thank the *C. elegans* Core Facility of the National Core Facility for Biopharmaceuticals, Ministry of Science and Technology. We thank WormBase for data support. Some strains were provided by the Caenorhabditis Genetics Center (CGC), which is funded by NIH Office of Research Infrastructure Programs (P40 OD010440) and some strains were provided by National BioResource Project (NBRP), which is funded by the Japanese government.

## Author Contributions

**Conceptualization:** Chih-Hsien Lin, Chan-Yen Ou.

**Funding acquisition:** Chan-Yen Ou.

**Investigation:** Chih-Hsien Lin, Ying-Chun Chen.

**Resources:** Shih-Peng Chan.

**Supervision:** Chan-Yen Ou.

**Writing – original draft:** Chih-Hsien Lin, Chan-Yen Ou.

**Writing – review & editing:** Chih-Hsien Lin, Ying-Chun Chen, Shih-Peng Chan, Chan-Yen Ou.

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
