## [Decision Letter · Decision Letter 0]

9 Mar 2022

Dear Dr Ou,

Thank you very much for submitting your Research Article entitled 'Multiple domains of TIAM-1 pattern axon and dendrite development through UNC-119 at the interface between actin and microtubule' to PLOS Genetics.

The manuscript was fully evaluated at the editorial level and by independent peer reviewers. The reviewers appreciated the attention to an important problem, but raised substantial concerns about the current manuscript. Based on the reviews, we will not be able to accept this version of the manuscript, but we would be willing to review a much-revised version. We cannot, of course, promise publication at that time.

Should you decide to revise the manuscript for further consideration here, your revisions should address the specific points made by each reviewer. We will also require a detailed list of your responses to the review comments and a description of the changes you have made in the manuscript. As editor, I remind you that our policy (below) is that a revised manuscript must include all of the numerical data underlying graphs or summary statistics; this can be provided as a supplemental file. And, I ask that graphs show individual data points for each genotype/condition; if there are too many data points for bar graphs, then other presentation formats should be used (violin plots, etc). Finally, the CGC requests specific information be included in acknowledgements; please check their website for details.

If you decide to revise the manuscript for further consideration at PLOS Genetics, please aim to resubmit within the next 60 days, unless it will take extra time to address the concerns of the reviewers, in which case we would appreciate an expected resubmission date by email to plosgenetics@plos.org.

[LINK]

We are sorry that we cannot be more positive about your manuscript at this stage. Please do not hesitate to contact us if you have any concerns or questions.

Yours sincerely,

Anne C. Hart

Associate Editor

PLOS Genetics

Gregory P. Copenhaver

Editor-in-Chief

PLOS Genetics

Reviewer's Responses to Questions

**Comments to the Authors:**

Reviewer #1: In the current Manuscript Lia and coworkers find that the actin regulator TIAM-1 has an interesting effect on PVD neuron development. In the tiam-1 mutant dendrite branching is reduced whereas the axon extends. They show that TIAM-1 interacts with a member of the cortical anchoring complex that connect the microtubule cytoskeleton to the cortex. In this way the regulation of the actin and microtubule cytoskeleton might be connected and may be differently regulated in the axon vs the dendrite.

This would be an interesting message, however I think that current manuscript does not connect enough TIAM role in actin (not shown except for branching) to its role on MTs (potentially via UN-119). More importantly, I cannot properly assess the experiments presented in this manuscript. For most of the quantifications presented, I cannot find how this was performed and what the numbers are. To give a few examples: F2G,L,M I have no idea what and how this was quantified. In L for example how come the motor does not accumulate in the axon in wildtype? Is this really what the authors see? Fig 5C-D How as this quantified? Did the authors trace the whole neuron? But then how did they work around the autofluorescence. Are these maximum projections or sum projections of the neuron Fig 6C,F where and how was the intensity quantified? Is it average intensity? Because I don’t really see it back in the figure.

Therefore at this stage I the first need to rewrite the manuscript to get a proper revision.

Still here are a few extra thoughts:

- The author studied the functions of TIAM-1 domains and conclude that the EVH1 domain is required for development of high order dendrites, and PDZ is responsible for primary dendrites, and the C-terminal region DHPH regulates the axon. Instead of quantifying total dendrites length (Figure2F), quantification of primary dendrite and sidebraches could be separated to better discriminate between actin rich neurite regions vs MT rich neurite region.

- B isoform seems to be much more important for general neuron development. However the precise contribution (vs A) would need a B specific mutant. Especially considering the very subtle defect of the A isoform. Rescues with the N-terminus is pretty subtle and I would draw too much conclusions based on this.

- Considering the strong morphology defect, quantifying % axon Mitochodria seems not correct to suggest a transport defect. Maybe this is just a consequence of the shorter dendrite and longer axon.

- “These expression patterns revealed that TIAM-1 localizes at the interface between MT and actin filaments.” and Figure 4: TIAM overlap with MT/actin; in neurites these will always overlap. Therefore not the best to prove connection. Maybe some life imaging would be nice to show co-movement. Alternatively try some actin depolymerization.

- Can unc-119 overexpression suppress tiam phenotypes?

- Where is the wildtype in 2F

- “Since UNC-119 mediates the link between MTs and actin filaments through UNC-33/CRMP and UNC-44/ankyrin in the cortical anchor complex “ that paper did not show any direct actin link.

- “The plus-end-directed Motor Kinesin-1 Prevents plus-end-out MTs from distribution in dendrites in C. elegans” – this is not the latest Shen lab model for kinesin-1

- Are the mutants use null mutants?

Reviewer #2: This manuscript by Lin et al investigates the role of the RacGEF TIAM-1 in controlling axon and dendrite development of the multi-dendritic PVD neurons of C. elegans. While others have investigated the role of this protein in axon and dendrite development, the strength of this paper is it’s finding of the opposite effects of a single gene on growth of the axon and dendrite in a single neuron. I found this manuscript to make a valuable contribution to our understanding of mechanisms controlling differential neurite development. I do suggest several changes below that I think will strengthen the manuscript. However, the only concerns that may require additional experimentation are items 1 and 2 below that will involve testing the functionality of the fluorescent reporter already made and an additional examination of isoform specific localization.

1) The authors report the localization of a TIAM-1::GFP reporter driven in the PVD neurons. However, I did not see an examination of the ability of this reporter to rescue any of the tiam-1 mutant phenotypes. Also, it wasn’t clear from the text or the supplemental table whether the cDNA used for this construct was the TIAM-1A isoform.

2) A major component of this story is the differential impact of TIAM-1 isoforms on axon vs dendrite development. The authors accomplished this through isoform-specific rescue. I was hoping to see whether the isoforms are differentially localized and if this may contribute to the different actions. Development of isoform-specific reporters would enhance interpretation.

3) The manuscript suffers from many grammatical errors that distract from an otherwise informative text. While many of the errors are minor and I could interpret the authors meaning, it created a more challenging read than necessary.

4) There were several instances where I disagree with the authors methods for data analysis or noted inconsistency in their analysis of the data.

a. It was unclear what level of ɑ the authors consider as statistically significant. For example, on page 11, the authors indicate that disruption of fhod-1 caused only a slight change in axonal or dendritic growth whereas the unc-60(su158) mutant had significant axonal overgrowth. However, both showed a statistical difference at p<0.05.

b. The isoform-specific rescue of axon length suggests a more nuanced interpretation than the authors give. While TIAM-1A and TIAM-1B provided full rescue, TIAM-1C only partially rescued the axon length. This could be due to the inherent variability in transgene expression. Alternatively, it could suggest that the EVH1 and/or the full PDZ domain are required. These alternatives are not recognized in the model of Fig 2H. Similarly, the authors state that the TIAM-1N539 construct restored dendritic development; however, this rescue appears slight compared with the full rescue shown by TIAM-1A and 1B.

c. I was appreciative of the authors use of relative dendritic length for many of the measurements—this appropriately recognized that the change in body length may make an absolute measurement of dendritic length misleading. I recommend they also consider relative length for the measurement of number of synaptic vesicles in the axon (Fig 3E).

d. Unlike the use of relative dendritic length, I was not enthusiastic about the authors use of relative fluorescence among cell compartments in Figure 5. This type of analysis could be misleading if the total fluorescence changes in only one compartment. For example, if the fluorescence gets much brighter in the axon, but stays the same in the dendrite this analysis will show a decrease in fluorescence in the dendrite even though there was no real change. Similarly the measurement of axon/dendrite ratio of UNC-119::mCherry and UNC-33::GFP does not discern whether the change in expression occurs in one compartment or both. Nor does it appear to account for relative length differences between WT and the mutant.

e. The interpretation of TIAM-1 being associated with actin and attaching to MT seemed a stretch. While the images certainly show localization in the same area, F-actin appeared to be so widely expressed throughout the neurites that a specific association is difficult to determine. The resolution of the Airyscan super-resolution microscope is approximately 150 nm laterally and 400 nm axially. This is much greater than the expected size of F-actin or RacGEF. Therefore, co-localization via light microscopy alone cannot be used to confirm a direct interaction.

5) Other minor issues:

a. The authors should clearly define what is meant by higher-order dendritic branches.

b. I assumed that the tiam-1(ok772) is a putative null. If so, this should be stated.

c. Line 142: The authors should quantify the number of high-order dendrites in the TIAM-1B rescue.

d. The introduction does not put the study in context of other C. elegans studies on TIAM-1. While the authors bring these up in the discussion, it would be valuable to include in the introduction.

e. Line 161: Provide citation for RAB-3 and TOMM-20 labeling synapses and mitochondria, respectively. The authors cite Knobel et al for the RAB-3 construct; however, I did not see mention of RAB-3 in this article.

f. Line 169: Provide citation for transport of synaptic vesicles and mitochondria by kinesins.

g. Line 173: Provide citation for microtubule orientation in PVDs.

h. Figure 4A. The authors state that TIAM-1::GFP was localized in the primary dendrite. Based on the image, it was difficult to determine if the posterior primary dendrite also contained this expression.

i. It may be beneficial to test the CRISPR-generated (ntu) mutants in the examination of UNC-119 expression. Is a specific isoform required for expression?

j. Line 258: It would be useful to request the Bulow lab CRISPR mutant lacking GEF activity to definitely test if this activity within the C-terminus is required for axon development.

k. Many of the images are missing scale bars. If all images in a single figure use the same scale, this should be stated in the legends.

l. The KLP-15::GFP signal is difficult to discern due to background autofluorescence.

Reviewer #3: The authors characterize the effects of different actin regulators on PVD axon and dendrite development in C. elegans. TIAM-1, a RacGEF, differentially regulates both axon and dendrite growth in vivo. TIAM-1's EVH1 and PDZ domains are required for dendrite development and its C-terminal DHPH domain is required only for axon development. In tiam-1 (ok772) mutants the distribution of actin and microtubules is altered. TIAM-1 is present in the same complex with UNC-119, and genetically interact to control axonal length/outgrowth.

Several concerns should be addressed so that data support the conclusions drawn before acceptance. Authors make claims of Tiam-1's role on the cytoskeleton and polarized cargo distribution. It is important that authors assess microtubule orientation, please exercise care in interpreting localization studies between TIAM-1 and cytoskeletal proteins and changes in organization of cytoskeleton markers. These experiments are the cornerstone on which interpretations of genetic interactions depend. Currently data as presented in these experiments do not support conclusions. More detailed comments are given below.

Major concerns

(1) In line 133, authors mention that the allele ok772 disrupts all the transcripts of tiam-1. Please provide experimental evidence/reference that shows that this allele disrupts all transcripts. This is important for the isoform-specific rescue experiments presented later in the study.

(2) In Fig 1 F, G, H the representative image wve-1(ok3308) shows a much longer total dendritic length as compared to tiam-1 (ok772). However in the accompanying graph in fig 1 H, tiam-1 (ok772) shows total dendritic length similar to wve-1(ok3308). It would be good to show representative images that reflect the data presented in the graph.

(3) In line 165,166,167 and data presented in Fig 3 A, B and E an increase in the number of SV clusters and changes in mitochondrial numbers could arise from changes in length of the neuronal processes, differences in levels on motors on cargo surface, altered retention etc. Authors present a very narrow interpretation that tiam-1 plays roles in polarized recruitment of cargo. It is unclear how authors exclude other interpretations of the cargo distribution data merely on the strength of the UNC-116::GFP and UNC-104::GFP motor distribution. It is important for authors to examine EB growth direction in PVD axons and dendrites in tiam-1 to determine if the polarized distribution of microtubules remains unchanged in tiam-1.

(4) Greater care needs to be taken in interpreting data presented in Figure 4, currently the data do not support the conclusions drawn from these experiments. The resolution of images presented do not allow one to conclude that Tiam-1 (stated in line 191) is present on the "inner surface of cortical actin bundles (labeled by RFP::UtrCH) (Fig 4B) while they also attached to MT bundles in the cell body". How do authors rule out that the signals ovelap just by chance and do not reflect close juxtaposition. In the case of actin, the actin bundles are present along narrow processes and any signal present in that region including for instance some cargo could well be present in areas that show actin signal. Likewise, in the cell body most Tiam-1 is not present closely associated with microtubules. Lines 192-194, Authors claim that TIAM-1 localizes to the basal region of actin-rich bundles, but in fig 4D, TIAM-1::GFP punctate are present in regions where RFP::UtrCH intensity is low. In the experiment with Ptrn-1, some quantitation where the number of puncta that are present adjacent to each other and the number that are present separately would be very useful to present. There is insufficient evidence from the mere co-localization between UNC-119 and Tiam-1 to support the claim that “TIAM-1 localises at the interface between MT and actin filaments”. The regions of UNC-119 enrichment could also be just changes in diameter of the process or accumulations unrelated to actin-microtubule interfaces. It would be helpful if authors alter the microtubule-actin junctions show an altered Tiam-1 localization.

(5) In figure 5, authors interpret the data of cytokeletal markers in the dendrite and primary dendrite as evidence for a necessary role for Tiam-1 in forming a boundary between the microtubule and actin cytoskeleton. It is unclear how authors conclude this from the images presented. There are changes in actin and microtubule cytoskeleton in the cell body of Tiam-1. The penetrance of these phenotypes and degree of disruption should be presented as a graph. The changes presented in the primary dendrite of tiam-1 could also arise from developmental changes where the processes are wider just an increase in actin and microtubules that their data do support. These conclusions should be revisited or alternate data to support authors claim should be presented.

(6) Authors claim that in tiam-1 mutant, UNC-119 distribution increases in the axon. Could this increase merely be a reflection of the increased axonal length? In Fig 6 A and B, tiam-1(ok772) mutant animals show UNC-119::mCherry localization in higher-order dendrites that is absent from wild type images. How does one account for this localisation?

(7) In Figure 6L authors see a doublet with anti-FLAG. Are there additional bands? Are these different isoforms of UNC-119? It would be useful if authors provide some explanation for this.

Minor concerns

(1) The paper identifies a genetic pathway for axon outgrowth, however there is nothing in the introduction about axon outgrowth and the roles of actin and microtubules in this process. This sets up the paper well for the differential roles of TIAM-1 that they describe. Such a paragraph would be very useful especially for a new student/post-doc in the field. I strongly suggest adding that.

(2) In all figure with graphs with error bars eg Figure 1 H and I do please list the n’s in the legend. It is more useful to present the data as violin plots so that the variability can be assessed.

(3) Please state how the act-3 gof allele is thought to act. Does it prevent attachment of other actin remodeling complexes or affect the kinetics?

(4) Line 142-143, Authors say that “TIAM-1B expression did not rescue distal high-order dendrite morphogenesis defects”. Graphs in Fig 2F suggest that overexpression of TIAM-1B in tiam-1(ok772) partially rescues the total dendritic length. It would be useful to explain this in their interpretation and also consider showing rescue in primary and higher order dendritic length in a supplementary figure.

(5) In Fig 6 D and E, several images of differing background intensities are stitched together to reconstruct the PVD neuron. It is unclear if such images were directly used to calculate the protein level ratios presented in the graphs. It would be useful to provide a more detailed description in the methods of their image processing pipeline leading upto the quantitation they present.

(6) Authors have used 'cargoes' three times (lines 50, 300, 309) and 'cargos' four times (lines 58, 169, 181, 327). Please use consistent spelling throughout the manuscript.

**Have all data underlying the figures and results presented in the manuscript been provided?**

Reviewer #1: **No: **there is nearly no info for any quantification, how this was performed and numbers

Reviewer #2: None

Reviewer #3: Yes

PLOS authors have the option to publish the peer review history of their article (what does this mean?). If published, this will include your full peer review and any attached files.

Reviewer #1: No

Reviewer #2: No

Reviewer #3: No

---

## [Decision Letter · Decision Letter 1]

14 Sep 2022

Dear Dr Ou,

Thank you very much for submitting your Research Article entitled 'TIAM-1 differentially regulates dendritic and axonal microtubule organization in patterning neuronal development through its multiple domains' to PLOS Genetics.

The manuscript was fully evaluated at the editorial level and by independent peer reviewers. The reviewers appreciated the attention to an important topic but identified a few minor concerns that we ask you to consider addressing in a revised manuscript. You can decide if you want to include or address these in a re-revised manuscript. Please note that we do not plan to send the re-revised manuscript to the external reviewers; I will review the minor changes, if any, as Academic Editor.

We therefore ask you to modify the manuscript according to the review recommendations. Your revisions should address the specific points made by each reviewer. 

1) Provide a detailed list of your responses to the review comments and a description of the changes you have made in the manuscript. If you change anything else in the manuscript, please also explain that in detail.

[LINK]

Yours sincerely,

Anne C. Hart

Academic Editor

PLOS Genetics

Gregory P. Copenhaver

Editor-in-Chief

PLOS Genetics

Reviewer's Responses to Questions

**Comments to the Authors:**

Reviewer #1: Overall I’m pretty happy with the extended methods and the extra data/answers to my other point. Also I think the EB analysis and focus on MTs strengthens the paper well.

I now mainly have minor points left that should easily be addressed.

- Fig S5 - I do not think that an UtrCH is a good marker to mark the cell cortex to define the distance of TIAM to the membrane. Especially in WT the actin filaments may not be at the cortex and thus the actual distance of TIAM to the membrane might be different. In actin depletion, I guess there is more cytosolic marker working as a fill and thus would work better. Therefore I’m not so sure about this statement line 234 “These results suggest that cortical actin bundles are required for TIAM-1 localization in cell cortex under cell membrane.”

- Fig 4 - Colocalization quantification – Is the whole CB used? If so for WT the main colocalization quantified would be the big blobs on left of the CB which for sure are not MTs. So this does not work.

Small changes

- UNC-119 comes out of the blue (line 235) – some introduction would be nice or move it to a later paragraph

- Line 297: “Based on our findings, we propose a working model that TIAM-1 organizes actin filaments and MTs to direct axonal/dendritic development by regulating UNC-119 (Fig 7H).” does this suggest that UNC-119 regulates actin? Based on what data?

- Figure 3D-E use of red/blue colors confusing. Blue is defective? Maybe turn colors around?

In general the manuscript would benefit from having a native speaker correcting the grammar. Here are a few textual suggestions to start with:

- Neurons do not really migrate (line 52); remove ”the” (line 118); remove “had” (line 126); “well MT organization” is not very nice (line210) maybe replace Well by Proper; Line 213 “We found that TIAM-1A::GFP specifically localized TO THE somatodendritic compartment with punctate patterns in THE primary and high-order dendrites but DOES not LOCALIZE TO the axon (Fig 4A).”; remove “Fig” line 244; replace “manifested” line 286 by e.g. elucidated; line 335 “Although THE MT”

Reviewer #2: The authors have satisfied my concerns in this revision.

Reviewer #3: The authors have resubmitted an excellent manuscript that has addressed all my concerns. I do think that the airy scan images are a good addition but still do not necessarily support the localization of TIAM-1 as being associated with actin and microtubule cytoskeleton. Using soluble GFP as a control may not be the best method. Could another microtubule or actin related protein have been used? I think no additional experiments are necessary however, it would be useful to be more careful in what is claimed for these data. An excellent paper and a valuable addition to the literature.

**Have all data underlying the figures and results presented in the manuscript been provided?**

Reviewer #1: None

Reviewer #2: Yes

Reviewer #3: Yes

PLOS authors have the option to publish the peer review history of their article (what does this mean?). If published, this will include your full peer review and any attached files.

Reviewer #1: No

Reviewer #2: No

Reviewer #3: No

---

## [Editor Report · Decision Letter 2]

29 Sep 2022

Dear Dr Ou,

We are pleased to inform you that your manuscript entitled "TIAM-1 differentially regulates dendritic and axonal microtubule organization in patterning neuronal development through its multiple domains" has been editorially accepted for publication in PLOS Genetics. Congratulations!

Yours sincerely,

Anne C. Hart

Academic Editor

PLOS Genetics

Gregory P. Copenhaver

Editor-in-Chief

PLOS Genetics

Comments from the reviewers (if applicable):

**Data Deposition**

http://datadryad.org/submit?journalID=pgenetics&manu=PGENETICS-D-22-00055R2

**Press Queries**

---

## [Editor Report · Acceptance letter]

6 Oct 2022

PGENETICS-D-22-00055R2 

TIAM-1 differentially regulates dendritic and axonal microtubule organization in patterning neuronal development through its multiple domains 

Dear Dr Ou, 

We are pleased to inform you that your manuscript entitled "TIAM-1 differentially regulates dendritic and axonal microtubule organization in patterning neuronal development through its multiple domains" has been formally accepted for publication in PLOS Genetics! Your manuscript is now with our production department and you will be notified of the publication date in due course.

With kind regards,

Agnes Pap

PLOS Genetics

On behalf of:
